# FEDERATED FEW-SHOT CLASS-INCREMENTAL LEARNING

**M. Anwar Ma'sum**[*]**, Mahardhika Pratama, Lin Liu, Habibullah Habibullah, and Ryszard Kowalczyk**

University of South Australia, Mawson Lakes, SA, 5095, Australia
masmy039@mymail.unisa.edu.au, dhika.pratama@unisa.edu.au, lin.liu@unisa.edu.au,
habibullah.habibullah@unisa.edu.au, ryszard.kowalczyk@unisa.edu.au

## ABSTRACT

This study proposes a challenging yet practical Federated Few-Shot Class-Incremental Learning (FFSCIL) problem, where clients only hold very few samples for new classes. We develop a novel Unified Optimized Prototype Prompt (UOPP) model to simultaneously handle catastrophic forgetting, over-fitting, and prototype bias in FFSCIL. UOPP utilizes task-wise prompt learning to mitigate task interference and over-fitting, unified static-dynamic prototypes to achieve a stability-plasticity balance, and adaptive dual heads for enhanced inferences. Dynamic prototypes represent new classes in the current few-shot task and are rectified to deal with prototype bias. Our comprehensive experimental results show that UOPP significantly outperforms state-of-the-art (SOTA) methods on three datasets with improvements up to 76% on average accuracy and 90% on harmonic mean accuracy respectively. Our extensive analysis shows UOPP robustness in various numbers of local clients and global rounds, low communication costs, and moderate running time. The source code of UOPP is publicly available at https://github.com/anwarmaxsum/FFSCIL.

## 1 INTRODUCTION

The previous studies on Federated Class Incremental Learning (FCIL) address catastrophic forgetting challenges in a dynamic environment with data privacy constraints. Coordinated by a central server, a collection of clients continually develops a global recognition model without sharing their local data. The first issue of FCIL is the existing works i.e. LGA (Dong et al., 2023), TARGET

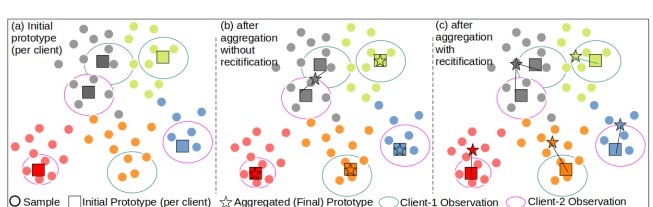

Figure 1: The importance of prototype rectification to handle prototype bias (a) Initial prototype per client (b) Aggregation without rectification can't handle prototype-bias (c) Aggregation with rectification overcomes prototype-bias.

(Zhang et al., 2023b) and LANDER (Tran et al., 2024) assume the clients carry abundant training data and thus impractical in the resource-constrained environments. They are data-hungry such that they face the issues of prototype bias and over-fitting in realm of the data scarcity constraint. As in stand-alone few-shot learning (Zhang et al., 2022), federated learning with few samples leads to prototype bias problems. Figure 1 visualizes how prototypes of the observed classes are generated and aggregated from the few samples carried by the clients. In FCIL simulation (Dong et al., 2023; 2022), where a client holds only partial classes e.g. 60% of the total classes, it shows that the observed few samples by a client lead to prototype bias where a prototype doesn't represent the true population rather it represents the few locally gathered samples. To overcome the prototype bias problem, each prototype should be refined to a correct location. As shown in Figure 1, we also emphasize that aggregating the prototype can't handle the prototype bias. Few-shot learning (FSL) e.g. MetaNode (Zhang et al., 2022) or few-shot class incremental learning (FSCIL) e.g. S3C (Kalla & Biswas, 2022) methods can't be expected since the methods require the presence of all classes whereas, in a federated setting, a client carries only a subset of all classes.

---

[*]Corresponding author

Second, the current FCIL methods train and share whole backbone parameters resulting in a large number of parameters during optimization processes which imply a long training time and a high communication cost. Third, in the current SOTAs, clients generate and share synthetic images for global model aggregation on the server side. Aside from overloading the communication costs, this mechanism may violate data privacy principles since synthetic data may reveal partial information about the private data of a client to another party due to its similarity to the original data. Last but not least, some FCIL methods e.g. LGA (Dong et al., 2023) GLFC (Dong et al., 2022) save several exemplars from previous tasks for rehearsal that may breach data openness policy, where data are only open for a client at a specific moment.

These gaps motivate us to address a new direction of FCIL i.e. Federated Few-Shot Class Incremental Learning (FFSCIL) where a client participating in the federated learning process only possesses very few samples. Second, we develop a novel efficient but effective approach to the FFSCIL problem, where a client trains and shares as small parameters as possible but produces a highly accurate global model without sharing any synthetic samples or saving exemplars from previous samples. Therefore, our proposed method handles catastrophic forgetting in dynamic collaborative learning with data privacy and data scarcity constraints. **The contributions of this paper are**: (1) We emphasize the data scarcity issue leading to the prototype-bias and over-fitting problems in FCIL and define a new problem, namely Federated Few-Shot Class Incremental Learning (FFSCIL); (2) We propose a **novel** method for the FFSCIL problem termed Unified Optimized Prototype Prompt (UOPP) built upon a prompt learning framework coupled with static and dynamic prototypes optimized by Neural Ordinary Differential Equation (ODE). The proposed method utilizes an adaptive dual-head to enhance its predictive accuracy. To our knowledge, our proposed method is the **first** prompt-based that integrates prompt tuning, prototype rectification by trainable network and adaptive dual classifiers in a single pipeline; (3) We offer theoretical studies for the convergence and generalization of the proposed method; (4) We provide a comprehensive analysis in three benchmark datasets that show the proposed method outperforms the baseline and current SOTAs with significant gaps and achieves improved stability-plasticity balances. Our analysis emphasizes the robustness of the proposed method in various participating clients and small rounds per task.

## 2 RELATED WORKS

**Federated Class Incremental Learning (FCIL):** The FCIL studies address catastrophic forgetting problem while preserving dta privacy e.g. FedWeIT (Yoon et al., 2021), GLFC (Dong et al., 2022), and LGA (Dong et al., 2023) optimize the global model by aggregating locally optimized models by the participating clients. The current SOTAs prove their effectiveness rather than combining FedAvg (McMahan et al., 2017) and class incremental learning (CIL) method such as ICARL(Rebuffi et al., 2017) and BiC (Wu et al., 2019). However, the SOTAs tune and send the whole backbone, producing long training times and high communication costs as a consequence. Furthermore, the SOTAs assume that a client saves several samples as memory that may not be practically applicable. Other studies i.e. TARGET (Zhang et al., 2023a) leverage synthetic samples for rehearsal instead of real samples. It achieves higher performance than FedWeIT (Yoon et al., 2021) but still outperformed by LGA. A different approach i.e. FedCIL (Qi et al., 2023) trains a generative model i.e. ACGAN (Odena et al., 2017) on local clients side to generate fake samples for aggregation on the central server's side. It achieves a higher performance than the combination of FedAvg or FedProx (Li et al., 2020) with ACGAN, DGR(Shin et al., 2017) or LWF-2T (Usmanova et al., 2021), but it needs more expensive communication costs and training time due to the generative model.

**Few Shot Class Incremental Learning (FSCIL):** Previous studies on FSCIL have attempted to maintain stability-plasticity tradeoff under data scarcity by adding extra representation e.g. TOPIC (Tao et al., 2020b) introduces Neural gas as the graph of mapped features and CEC (Zhang et al., 2021) continually evolves its classifier to adapt to new tasks. Another approach modifies its learning mechanism e.g. FSLL (Mazumder et al., 2021) updates with self-supervised loss, F2M (Shi et al., 2021) finds flat minima regions on the base task then forces parameter updates on few shot tasks to reside within the flat region, S3C (Kalla & Biswas, 2022) trains scholastic classifier with supervised loss and MgSvF (Zhao et al., 2024) applies multi grained fast-slow learning mechanism. Prototype-based methods e.g. TEEN(Wang et al., 2024), NC-FSCIL(Yang et al., 2023), and OrCO(Ahmed et al., 2024) show the important of prototype correction to deal with prototype bias in FSCIL. However, prototype refinement in the data scarcity is still an open challenge. Besides, FSCIL methods aren't yet proven in federated settings under non-i.i.d constrain. A comprehensive literature review is presented in Appendix G.

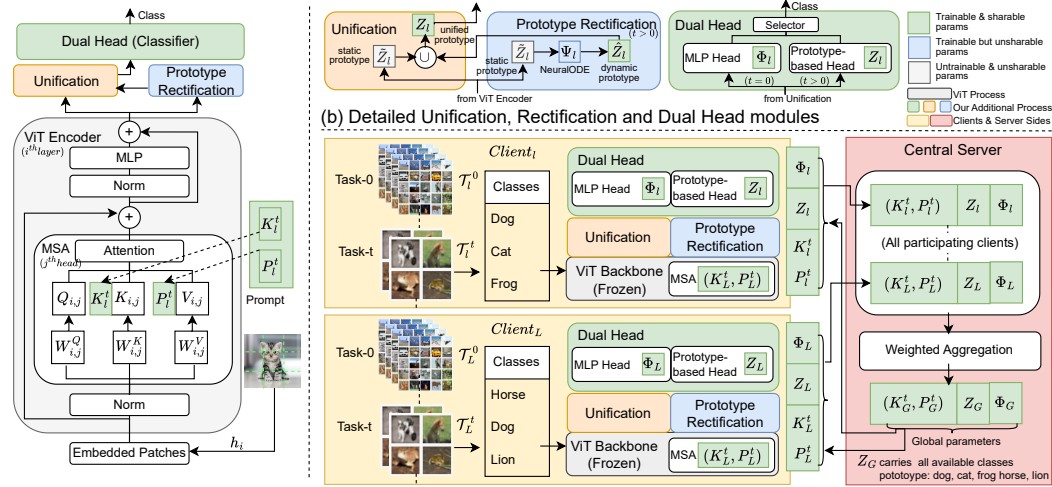

Figure 2: The visualization of UOPP, includes task-wise prompt learning empowered by shared unified static-dynamic prototypes, dynamic prototypes rectification, adaptive dual heads, and weighted aggregation. The gray-colored parameters are frozen and unshareable parameters, the green-colored parameters are trainable and shareable parameters, while the blue-colored parameters are trainable but unshareable parameters

## 3 PROBLEM FORMULATION

**Federated Few-Shot Class-Incremental Learning (FFSCIL)** is defined as: Given a sequence of tasks $[0, 1, 2, ..., T]$ where each task $t$ carries a labeled training set $\mathcal{T}^t = \{(x_i^t, y_i^t)\}_{i=1}^{|\mathcal{T}^t|}$, where $x_i^t \in X$ denotes an input image and $y_i^t \in Y$ denotes its label, and $|.|$ denotes the cardinality. Each task $t$ is disjoint with another task $t'$. i.e. $\forall_{t,t'} \mathcal{T}^t \cap \mathcal{T}^{t'} = \emptyset$. On each task-$t$, a set of clients $\{l\}_{l=1}^{L_{all}}$ coordinated by a central server $G$ are deployed to learn $\mathcal{T}^t$. In the first task ($t = 0$), each client $l$ carries abundant training samples while the remaining tasks ($t > 0$), it carries far smaller samples than the first task i.e $|\mathcal{T}_l^0| >> |\mathcal{T}_l^{t>0}|$. For a convenient way, task-0 is called the base task while the rest is called the few-shot task (FS task). A client $l$ holds only a subset of current task training set i.e. $\mathcal{T}_l^t \subset \mathcal{T}^t$. Each client-$l$ carries non-identically and distributed data (non-i.i.d) to another client $l'$ i.e. $\forall_{l,l'} \mathcal{D}_l^t \neq \mathcal{D}_{l'}^t$ where $\mathcal{D}_l^t$ and $\mathcal{D}_{l'}^t$ are distribution of $\mathcal{T}_l^t$ and $\mathcal{T}_{l'}^t$ respectively. Following FCIL, Non-i.i.d distribution is represented by the percentage of available classes $\eta$. Each task-$t$ is learned in a federated way that is repeated in $R_T$ rounds where in each round $r \in [1..R_T]$, a set of local clients is randomly selected from all available clients i.e. $\{l\}_{l=1}^L \subset \{l\}_{l=1}^{L_{all}}$. Due to data privacy constraints, a client-$l$ is not allowed to share any training sample $(x_i, y_i) \in \mathcal{T}_l^t$ to another client or server, but permitted to share its parameters.

Let a deep neural network $g_\Phi(f_\Theta(.))$ be parameterized by $\Theta$ and $\Phi$ where $f(.)$ and $g(.)$ are the feature extractor and classifier respectively. In each round $r$ of task $t$, a central server $G$ coordinates selected local clients $\{l\}_{l=1}^L$ to conduct local CIL training using its training samples $\{\mathcal{T}_l^t\}$. Each client-$l$ optimizes its local parameters $(\Theta_l, \Phi_l)$, then sends its locally optimized parameters to the central server $G$ to be aggregated. The central server $G$ aggregates all received local parameters into optimum global parameters i.e. $(\Theta_G, \Phi_G) = Agg(\{(\Theta_G, \Phi_G)\}_{l=1}^L)$ and communicates them back to all clients for the next round process. **The objective of FFSCIL** is to achieve an optimum global model $g_{\Phi_G}(f_{\Theta_G}(.))$ to recognize the learned classes from the first task (task-0) until the current task (task-$t$) i.e. $\{\mathcal{T}^0, ..., \mathcal{T}^t\}$.

## 4 PROPOSED METHOD: UNIFIED OPTIMIZED PROTOTYPE PROMPT(UOPP)

We design our method, termed Unified Optimized Prototype Prompt (UOPP), to address challenges in FFSCIL i.e. catastrophic forgetting, over-fitting and prototype bias simultaneously under data privacy constraints. Figure 2 exhibits the flow of our method in both the local training view (a) and federated training view (b). Looking at Figure 2 (a), we utilize a prompt-based approach on top of the frozen ViT backbone as it minimizes task interference that leads to better handling of catastrophic forgetting with lightweight learnable parameters (prompts). Then we add a rectification block to handle prototype bias in the few-shot tasks ($t > 0$) by iteratively rectifying the prototypes.

Note that each client only holds few samples for any class in the few-shot tasks. Then we add a unification block to unify the rectified prototypes and the feature produced by the ViT encoder. Since the prototypes are shareable to/from the server, this mechanism handles a non-i.i.d challenge where a client-$l$ is accommodated to learn knowledge representation from the classes not available in $\mathcal{T}_l^t$. Last, we design an adaptive dual head that leverages the strength of both MLP and prototype-based classifiers. On the federated view (b), the Figure shows that in our method, a client shares only small-sized parameters i.e. prompts, prototypes, and MLP parameters rather than the whole backbone parameters. Thus, it minimizes the communication cost between clients and the central server. The details of our method are presented in the following subsections. The **uniqueness** of our method from the existing prototype-based methods i.e. (Wang et al., 2024), (Yang et al., 2023), (Ahmed et al., 2024), (Goswami et al., 2024), and (Guo et al., 2024) is that we utilize a trainable Neural ODE that works by support and query samples drawn from different distribution, while the existing method utilizes similarity ratio for rectification process. Second, alongside prototype rectification, we adjust task-wise key and prompt parameters to improve inter-task separability. Different from PILoRA (Guo et al., 2024), the prompt is prepended to $W^K x$ and $W^V x$ of ViT model, while PILoRA appends matrices $A.B$ into $W^Q$ and $W^V$ and doesn't use task-wise keys. Third, we utilize dual-head classifiers to leverage the strength of both combined with task prediction.

## 4.1 MINIMIZING TASK INTERFERENCE VIA PROMPT LEARNING

**Prompt Structure:** we define a prompt $\mathbf{P}_l$ for each client-$l$ to learn a sequence of task $\{t\}_{t=0}^T$ as: $\mathbf{P}_l = [(K_l^0, P_l^0), (K_l^t, P_l^t), .., (K_l^T, P_l^T)], t \in [0..T]$ where $(K_l^t, P_l^t)$ is a pair of prompt key-and-value corresponded to task-$t$. Our method utilizes the prefix tuning technique (Li & Liang, 2021) because it usually outperforms the prompt tuning method, then following the definition of prefix tuning, and attention mechanism in ViT, the output of ViT backbone given an input $x$ and prompt $(K_l^t, P_l^t)$ is defined as:

$$f_{(K_l^t, P_l^t)}(x) = A(Q_{ij}, [K_l^t + K_{i,j}], [P_l^t + V_{i,j}]) \tag{1}$$

where $Q_{i,j}, K_{i,j}, V_{i,j}$ are query, key and value of $j^{th}$ head MSA in $i^{th}$ layer of ViT encoder, $+$ denotes concatenation, and $A$ denotes attention function. Note that the function is applied for all MSA heads $j \in [1..J]$ and all encoder layers $i \in [1..I]$. At task-$t$, a client-$l$ optimizes only $(K_l^t, P_l^t)$ and not the other prompt key-value pair, and $(K_l^t, P_l^t)$ is adjusted only in task $t$, not in the previous or upcoming tasks. Therefore, once optimal to $\mathcal{T}_l^t$, the pair $(K_l^t, P_l^t)$ will remain robust against forgetting since its value is not adjusted afterward. In addition, $K_l^t$ is adjusted to match sample $x_i \in \mathcal{T}_l^t$ by using a matching loss $\mathcal{L}_m$ during training. This mechanism is designed to make $(K_l^t, P_l^t)$ exclusive to task-$t$ and not the samples from other tasks. In other words, it minimizes task interference. In addition, since only prompts rather than the whole network parameters are adjusted, this strategy alleviates the over-fitting problem due to few samples.

## 4.2 UNIFIED STATIC-DYNAMIC PROTOTYPE AND ITS USABILITY

**Static Prototype:** we define a vector of $D$-dimension $\tilde{z}_c$ as the prototype for class $c \in \mathcal{T}^t$ that is produced from ViT encoder, where $D$ is the embedding dimension. A static prototype set $\tilde{Z}_l = \{\tilde{z}_{lc}\}$ is a collection of static prototypes of class-$c$ that are available at client-$l$. Assuming that a prototype follows a Gaussian distribution i.e. $z_c \sim \mathcal{N}(\mu_c, \Sigma_c)$ and forms $D$ disjoint uni-variate distribution, then a prototype of class $c$ is represented as $z_c \sim \mathcal{N}(\mu_c, \sigma_c^2)$ where $\sigma_c^2 = I_D.\sigma_{c,i}^2, i \in \{1, 2, ..., D\}$, $I_D$ is identity matrix. Note that at task-$t$, a selected local client-$l$ holds its local training set $\mathcal{T}_l^t$. Suppose that $\mathcal{T}_{lc}^t = \{(x_{li}^t, y_{li}^t) \in \mathcal{T}_l^t, y_{li}^t = c\}$ is the samples of class-$c$ in $\mathcal{T}_l^t$ and $|.|$ denotes the number of samples. Then, a static prototype $\tilde{z}_{lc} \sim \mathcal{N}(\mu_{lc}, \sigma_{lc}^2)$ for a class-$c$ available in $\mathcal{T}_l^t$ is computed by Eq. 2 and 3.

$$\mu_{lc} = \frac{1}{|\mathcal{T}_{lc}^t|} \sum_{i=1}^{|\mathcal{T}_{lc}^t|} f_{(K_l^t, P_l^t)}(x_i), x_i \in \mathcal{T}_{lc}^t \tag{2}$$

$$\sigma_{lc}^2 = \frac{1}{|\mathcal{T}_{lc}^t|} \sum_{i=1}^{|\mathcal{T}_{lc}^t|} (\mu_{lc}^t - f_{(K_l^t, P_l^t)}(x_i))^2, x_i \in \mathcal{T}_{lc}^t \tag{3}$$

**Dynamic Prototype:** we define a vector $\hat{z}_c$ of $D$ dimension as the prototype for class $c \in \mathcal{T}^{t>0}$ iteratively rectified during local training on task-$t$, where $D$ is the embedding dimension. A dynamic prototype set $\hat{Z}_l = \{\hat{z}_{lc}\}$ is a collection of dynamic prototypes of class-$c$ available at client-$l$. At the beginning of local training at task $t > 0$ on client-$l$, a dynamic prototype $\hat{z}_{lc}$ is set by its corresponding static prototype value $\tilde{z}_{lc}$, then iteratively rectified by GradNet $g_{\Psi_l}$ during local training

i.e. $\hat{z}_{lc} = Rectification(g_{\Psi_l}, \tilde{z}_{lc})$. After finishing the training of task-$t$, the dynamic prototypes of all classes $c \in \mathcal{T}^t$ are not updated anymore and are stored as static prototypes. Therefore, the dynamic prototype set $\hat{Z}_l$ includes prototypes from currently learned classes only, while the static prototype set $\tilde{Z}_l$ includes both currently learned classes and previously learned classes. The details of the rectification process are explained in the following sub-section.

**Unified Prototype:** we define the unified prototype $Z_l = \tilde{Z}_l \cup \hat{Z}$ as the union of static and dynamic prototype of client-$l$. Following the definition of static and dynamic prototypes, $Z_l$ contains static prototypes of previously learned classes and dynamic prototypes of currently learned classes that are still updated during the local training at client-$l$. Unified prototype $Z_l$ plays an important role in our method, both in the base task ($t = 0$) and the few-shot task ($t > 0$). As illustrated by the unification block in Figure 2, in the base task, $Z_l$ is unified with the output of ViT i.e. $f_{(K_l^t, P_l^t)}(x)$. Since $Z_l$ is shareable, then $Z_l$ contains prototypes of classes unavailable in $\mathcal{T}_l^t$ shared by the central server. Therefore, each client-$l$ can afford to learn all classes in $\mathcal{T}^t$. In the few-shot task, each client-$l$ rectifies $\hat{z}_{lc}$ for all classes $c \in \mathcal{T}_l^t$. Similarly, the shareable $Z_l$ enhances the separability of $\hat{z}_{lc}$ since it is contrasted to all other prototypes of all learned classes.

### 4.3 HANDLING PROTOTYPE-BIAS VIA DYNAMIC PROTOTYPE RECTIFICATION

Previous studies (Chen et al., 2018; Zhang et al., 2022) emphasize that generations of prototypes by averaging few samples leads to prototype bias, as the prototypes only represent the observed samples but not the whole population, as illustrated in Figure 3(a). Therefore, our method rectifies a prototype $\hat{z}_c$ to become as close as possible to the population mean. We follow the framework

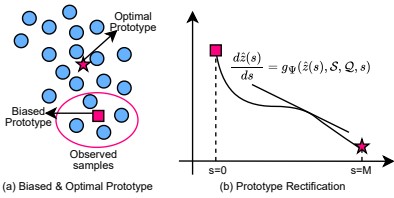

Figure 3: Visualization of Biased and Optimal Prototype (a), and Rectification (b)

of episodic training in few-shot learning (Vinyals et al., 2016; Zhang et al., 2022), a rectification process of a prototype $\hat{z}_c$ can be defined as: $\hat{z}_c(M) = Rectification(g_\Psi(), \hat{z}_c(0), \mathcal{S}, \mathcal{Q}, s = M)$ where $s$ is the number of rectification steps, $\hat{z}_c(0)$ and $\hat{z}_c(M)$ are the initial prototype and final prototype after $M$-steps rectification process, $\mathcal{S}$ is the support set that contains few observed samples per class, $\mathcal{Q}$ is the query set comprising unlabeled samples, and $g_\Psi()$ is gradient networks (GradNet) parameterized by $\Psi$. Now looking into a more detailed view, suppose that $L(\hat{z}_c(s))$ as a differentiable loss function with prototype $\hat{z}_c(s)$ and $\nabla L(\hat{z}_c(s))$ as its gradient during the optimization process, one step prototype rectification can be defined as an iterative process of Gradient Descent algorithm as described in equation 4. Symbol $\alpha$ denotes the learning rate, while $\omega$ denotes $l_2$-norm regularizer.

$$\hat{z}_c(s+1) = \hat{z}_c(s) - \alpha(\nabla L(\hat{z}_c(s)) + \omega(\hat{z}_c(s))) \tag{4}$$

The previous study (Chen et al., 2018) discovered that the iterative process above can be viewed as Euler discretization of an Ordinary Differential Equation (ODE). Therefore, the term $\nabla L(\hat{z}_c(s))$ in the process above can be derived into equation 5, where $s$ represents a continuous variable such as time, and $\frac{d\hat{z}_c(s)}{ds}$ represents a continuous gradient flow of prototype $\hat{z}_c^t(s)$ over $s$. Since the optimization process is executed by a neural network model $g_\Psi(.)$ (last part of equation 5), then ODE becomes Neural ODE.

$$-\nabla L(\hat{z}_c(s)) = \frac{d\hat{z}_c(s)}{ds} = g_{\Psi_l}((\hat{z}_c(s), \mathcal{S}, \mathcal{Q}, s)) \tag{5}$$

We follow the implementation of GradNet (Zhang et al., 2022) as the neural network model $g_\psi(.)$ that executes the rectification process. Following the implementation on GradNet, the optimum prototype is produced in the last step of rectification i.e. $\hat{z}_c(M) = \hat{z}_c(0) + \int_{s=0}^{M} g_{\Psi_l}((\hat{z}_c(0), \mathcal{S}, \mathcal{Q}, s))$. We follow the implementation of the previous study (Chen et al., 2018; Zhang et al., 2022), where the last term is solved by ODESolver based on Runge-Kutta method (Alexander, 1990). Therefore, the optimum prototype $\hat{z}_c(M)$ can be obtained by executing equation 6.

$$\hat{z}_c(M) = ODESolver(g_{\Psi_l}, \hat{z}_c(0), \mathcal{S}, \mathcal{Q}, s = M) \tag{6}$$

In previous studies (Chen et al., 2018; Zhang et al., 2022), $\mathcal{Q}$ contains images from base task. Now dealing with limitations in FFSCIL, we can't afford to save exemplars from previous tasks or gather images from other clients. However, we have a shareable unified prototype $Z_l = \tilde{Z}_l \cup \hat{Z}_l$ containing all learned class prototypes. Therefore, the **uniqueness** of our rectification is that we generate

support set $\mathcal{S}$ is constructed by drawing prototypes from $\mathcal{N}(\mu_{lc}, \sigma_{lc}{}^2)$ where $(\mu_{lc}, \sigma_{lc}{}^2)$ is computed by eq. 2-3 for all class $c$ available in $\mathcal{T}_l^t$, and generate query set $\mathcal{Q}$ by drawing prototypes from $\mathcal{N}(\hat{\mu}_l, \hat{\sigma}_l^2)$ where $(\hat{\mu}_l, \hat{\sigma}_l^2)$ is the property of $\hat{z}_l \in \hat{Z}_l$. As the implications, our rectification occurs fully in the embedding space and pseudo-rehearsal free. Note that $Z_l$ is assigned by aggregated prototypes i.e. $Z_l = Z_G$ in each federated round.

## 4.4 ADAPTIVE DUAL-HEAD CLASSIFIER FOR A BETTER PREDICTION

We design an adaptive dual head classifier by leveraging MLP classifier $g_\Phi(.)$ that works effectively in the highly labeled task ($\mathcal{T}^0$) and prototype-based (PB) classifier $g_Z(.)$ that has been proven robust in few shot tasks ($\mathcal{T}^{t>0}$). We combine both MLP and PB classifiers as one united head layer. MLP parameter $\Phi$ is optimized in the base task, while unified prototype set $Z$ is optimized in all few shot tasks. In the testing phase, we deploy a head selector to select which classifier to use for an input $x$. The head selector works by predicting the task ID based on the input-key matching. Given an input image $x$, and a sequence of prompt-key $K^0, K^1, ..., K^T$, we can find the task-id where $x$ belong by finding the highest similarity of $x$ and $K^t, t \in [0, ..T]$. The predicted label $\hat{y}$ of an input $x$ is computed by equation 7.

$$\hat{y} = \begin{cases} g_\Phi(f_{K^{(0}, P^0)}(x)), & \text{if } \hat{t} = 0 \\ g_Z(f_{(K^{\hat{t}}, P^{\hat{t}})}(x), & \text{otherwise}, \end{cases}, where, \; \hat{t} = arg\min_t \mathcal{L}_m(x, K^t) \tag{7}$$

where $\mathcal{L}_m$ is matching loss between input $x$ and prompt-key $K^t$. Note that equation 7 is applicable both for the client and server side. The united classifier is expected to elevate the prediction result, rather than employing one of the FC or PB classifiers alone, this hypothesis will be proven in our ablation study.

## 4.5 FEDERATED LEARNING AND SERVER-SIDE AGGREGATION

Figure 2 shows that on a task-$t$, each client-$l$ optimizes its local parameters based on its local training data $\mathcal{T}_l^t$. Then the optimized parameters i.e. $((K_l^t, P_l^t), \Phi_l, Z_l)$ to the central server $G$ are aggregated. We propose a simple weighted aggregation applying the principle "the more you learn, the better your knowledge". We consider the total participation of a client $S_l$ on the task $t$ as the client's weight $w_l$. In a round $r$ of a task $t$, given clients that carry locally optimized parameter $\{((K_l^t, P_l^t), \Phi_l, Z_l)\}_{l=1}^L$ and their weights $\{w_l\}_{l=1}^L$, then the aggregated parameter $((K_G^t, P_G^t), \Phi_G, Z_G)$ is computed using equation 8.

$$((K_G^t, P_G^t), \Phi_G, Z_G) = \frac{1}{\sum_{l=1}^L w_l} \sum_{l=1}^L ((K_l^t, P_l^t), \Phi_l, Z_l).w_l \tag{8}$$

In the base task ($t = 0$), the server aggregates and distributes prompt parameter $(K_G^t, P_G^t)$, MLP head parameter $\Phi_G$, and unified prototype set $Z_G^t$ to all participating clients. while in the few shot tasks ($t > 0$), the server aggregates and distributes prompt parameters and unified prototype set only. The GradNet parameter $\Psi_l$ is unshared and utilized only for local prototype rectification. Rather, a client sends the rectified dynamic prototype $\hat{Z}_l \subseteq Z_l$ to the central server.

## 4.6 WRAP UP AND FINAL OBJECTIVE

In the base task, each client-$l$ unifies $f_{(K_l^0, P_l^0)}(x)$ with $Z_l$ then updates prompt and MLP head parameters i.e. $(K_l^0, P_l^0)$ and $\Phi_l$. In the few-shot tasks, client-$l$ performs $f_{(K_l^t, P_l^t)}(x)$ to form static prorotype $\tilde{Z}_{lc}$, then construct support sample ($\mathcal{S}$) and query samples ($\mathcal{Q}$), and perform rectification for all $\hat{z}_{lc} \in \hat{Z}_{lc}$. Afterward, it updates prompt parameters $(K_l^t, P_l^t)$ and GradNet parameter $\Psi_l$. Therefore, the objectives of local training are defined in the equations below:

**(i). Base Task ($t = 0$) objective :**
$$\mathcal{L}_{t=0} = \mathcal{L}_{ce}(g_{\Phi_l}(f_{(K_l^0, P_l^0)}(x_i) \cup Z_l), y_i \cup \mathcal{C}_l) + \lambda \mathcal{L}_m(x_i, K_l^0), (x_i, y_i) \in \mathcal{T}_l^0 \tag{9}$$

**(ii). Few Shot Tasks ($t \geq 1$) objective:**
$$\mathcal{L}_{t>0} = \mathcal{L}_{pce}(g_{Z_l}(g_{\Psi_l}(\hat{Z}_{lc}(s), \mathcal{S}, \mathcal{Q}, s), \mathcal{C}_l) + \lambda \mathcal{L}_m(x_i, K_l^t), (x_i, y_i) \in \mathcal{T}_l^t \tag{10}$$

where $\mathcal{L}_{ce}$ denotes cross-entropy loss, $\mathcal{L}_{pce}$ denotes prototype cross-entropy loss utilizing cosine distance as similarity measurement, $\mathcal{L}_m$ denotes matching loss, and $\mathcal{C}_l$ is the label set of the unified prototype set $Z_l$, $(f_{(K_l^t, P_l^t)})$ denotes prompting output as in eq. 1, and $g_{\Psi_l}$ denotes rectification function as in eq. 5. The detailed process of our method is presented in Appendix A.

## 5 THEORETICAL ANALYSIS

Let $\Theta = (P, \Phi, \Psi)$ be trainable parameters, $F(\Theta) = \mathbb{E}[\mathcal{L}(\mathcal{T}; \Theta)] = \mathbb{E}[\mathcal{L}(\mathcal{T}; (P, \Phi, \Psi))]$ is the expected loss function, $k, E, R$, and $L_S$ is local iteration, local epoch, global round, and number of selected local clients respectively. We follow $L$-smooth and $\mu$-strongly convex $F$, $G$-bounded uniformly gradient assumptions, random uniformly distributed batches, and decreasing learning rate as in (Li et al., 2019),(Bottou et al., 2018) We stated **Theorem 1,2, and 3** as presented in Appendix B. Theorem 1 and 2 prove UOPP local training and federated convergence respectively, while theorem 3 proves UOPP generalization.

## 6 EXPERIMENTAL RESULTS AND ANALYSIS

### 6.1 EXPERIMENTAL SETTING

**Datasets:** our experiment is done using three benchmarks i.e. split CIFAR100, split MiniImageNet, and split CUB200. The CIFAR100 and miniImageNet datasets contain 100 classes while CUB200 is a dataset of 200 classes. We follow the settings from (Tao et al., 2020a) for tasks and classes per task split, and (Dong et al., 2023) for the federated setting. For CIFAR100 and MiniImageNet, we split the dataset into 9 tasks i.e. 60 classes for the base task ($t = 0$), and 5 classes for each few-shot task ($t \geq 1$). We split the CUB200 dataset into 11 tasks i.e. 100 classes for the base task, and 10 classes for each few-shot task. Few shot tasks are measured in 5-shot and 1-shot settings.

**Benchmark Algorithms:** UOPP are compared with 9 benchmark algorithms i.e. LGA (Dong et al., 2023), TARGET (Zhang et al., 2023b), LANDER (Tran et al., 2024), and Fed-CPrompt (Bagwe et al., 2023) that represent the SOTA in federated class incremental learning, Fed-S3C(Kalla & Biswas, 2022) that represents SOTA in few shot class incremental learning, Fed-L2P(Wang et al., 2022b), Fed-DualP(Wang et al., 2022a), Fed-CODAP(Smith et al., 2023), and PILoRA(Guo et al., 2024) that represent SOTA in FCIL. Except for Fed-CPrompt, "Fed-" denotes that the method is customized in a federated manner from its original (stand-alone) mode, by using FedAvg as the aggregation function. We only run LANDER and PILoRA for CIFAR100 since their official code, setting, and pre-trained embedding that can be executed in our setting is only for CIFAR100 dataset. We also evaluated our method in standalone FSCIL and compared it to FSCIL SOTAs i.e. TEEN(Wang et al., 2024), NC-FSCIL(Yang et al., 2023), OrCo(Ahmed et al., 2024), and PriViLege(Park et al., 2024), Please see Appendix D.1.

**Details and Metrics:** our numerical study is executed under a single NVIDIA A100 GPU with 40 GB memory across 3 different random seeds. Adapted from (Dong et al., 2023), the simulation is run by 20 total clients and 1 global server, where in each round, 6 (30%) local clients are selected randomly. Each client randomly receives 60% ($\eta = 0.6$) classes. The total global round is set to 90 (10 rounds/task) for CIFAR100 and MiniImageNet and 110 for CUB200. Our task split setting is different from the recent study (Jiang et al., 2024), since it follows FCIL setting, while our setting follows FSCIL setting.We evaluate the consolidated algorithms for all learned classes with accuracy metrics (Acc.) adn performance drop (PD). Besides, we measure the accuracy of base classes, novel classes, and harmonic mean accuracy that represents stability-plasticity performance. Please see Appendix F for the detailed experiment settings, hyperparameters, and metrics.

### 6.2 MAIN RESULTS

**a) General Performance:** the numerical result of the consolidated algorithms is shown in table 1. The proposed method (UOPP) achieves the highest accuracy with a significant gap $5 - 76\%$ compared to the competitor methods both in 5-shot and 1-shot settings. Fed-S3C, TARGET, LGA, and LANDER achieve relatively low performance with $30 - 76\%$ gap in 3 benchmark datasets in both 5-shot and 1-shot settings compared to UOPP. The results confirm that the FCIL and FSCIL methods can't be applied directly on FFSCIL. Meanwhile, prompt-based methods i.e. Fed-L2P, Fed-DualP, Fed-CODAP, and Fed-CPrompt achieve a relatively better performance than those 4 methods. Compared to UOPP, the methods have relatively smaller gap i.e. $5 - 36\%$. The results prove that prompt-based methods are more promising than the SOTAs of FCIL and FSCIL, The proposed method also achieves the lowest performance drop with $(0.7 - 19\%)$ followed by Fed-L2P with $(0.4-22\%)$ PD and Fed-DualP with $(-0.4-28\%)$ PD. Despite utilizing ViT and prototype approach, PILoRA achieves lower performance than the prompt-based method i.e. $< 52\%$ in average. Looking at per-task performance, Figure 4 shows that UOPP achieves the highest accuracy in all tasks in those

Table 1: Numerical result of the consolidated algorithms in CIFAR100, MiniImageNet, and CUB200 dataset in 5-shot and 1-shot setting across 3 different seeded runs.

(a) Complete numerical result on CIFAR100 dataset with 5-shot setting

| Method | Trainable Params. | Accuracy in each session (%) ↑ | | | | | | | | | Avg↑ | PD↓ | Gap↓ |
|---|---|---|---|---|---|---|---|---|---|---|---|---|---|
| | | 0 | 1 | 2 | 3 | 4 | 5 | 6 | 7 | 8 | | | |
| Fed-S3C | CNN | 44.51 | 48.97 | 47.77 | 45.35 | 43.48 | 41.47 | 40.33 | 39.32 | 37.71 | 43.21 | 6.80 | 46.80 |
| TARGET | CNN | 68.90 | 63.61 | 59.06 | 55.12 | 51.68 | 48.64 | 45.94 | 43.52 | 41.34 | 53.09 | 27.56 | 36.92 |
| LGA | CNN | 73.76 | 69.80 | 65.59 | 60.26 | 56.87 | 52.94 | 50.66 | 47.69 | 44.89 | 58.05 | 28.87 | 31.96 |
| LANDER | CNN | 58.60 | 61.75 | 56.26 | 52.11 | 47.71 | 44.71 | 41.69 | 40.28 | 38.87 | 49.11 | 19.73 | 40.90 |
| Fed-L2P | Prompt | 73.47 | 74.20 | 73.37 | 71.88 | 70.85 | 70.72 | 69.28 | 68.66 | 68.37 | 71.20 | 5.10 | 18.81 |
| Fed-DualP | Prompt | 76.39 | 82.75 | 83.37 | 80.80 | 79.93 | 78.26 | 77.73 | 76.98 | 77.11 | 79.26 | -0.72 | 10.75 |
| Fed-CODAP | Prompt | 81.73 | 69.29 | 70.81 | 68.67 | 67.17 | 66.14 | 64.32 | 64.79 | 64.12 | 68.56 | 17.62 | 21.45 |
| Fed-CPrompt | Prompt | 88.00 | 64.63 | 69.30 | 67.39 | 63.39 | 62.33 | 61.11 | 59.78 | 59.00 | 66.10 | 29.00 | 23.91 |
| PILoRA | Params (A,B) | 67.40 | 62.22 | 57.77 | 53.92 | 50.55 | 47.58 | 44.93 | 42.57 | 40.44 | 51.93 | 26.96 | 38.08 |
| **UOPP (Ours)** | **Prompt** | **90.57** | **90.58** | **90.85** | **90.96** | **91.23** | **91.51** | **91.56** | **91.74** | **81.05** | **90.01** | **9.52** | **0.00** |

(b) Summarized numerical result on CIFAR100 dataset with 1-shot setting, MiniImagenet, and CUB200 dataset with 5-shot and 1-shot settings

| Method | CIFAR100 | | | | | | MiniImageNet | | | | | | CUB200 | | | | | |
|---|---|---|---|---|---|---|---|---|---|---|---|---|---|---|---|---|---|---|
| | 1-shot | | | 5-shot | | | 1-shot | | | 5-shot | | | 5-shot | | | 1-shot | | |
| | Avg | PD | Gap | Avg | PD | Gap | Avg | PD | Gap | Avg | PD | Gap | Avg | PD | Gap | Avg | PD | Gap |
| Fed-S3C | 41.97 | 9.07 | 46.65 | 29.73 | 5.48 | 63.19 | 29.16 | 8.24 | 63.05 | 14.91 | 7.22 | 65.89 | 14.40 | 7.85 | 62.33 |
| TARGET | 53.09 | 27.56 | 35.53 | 44.77 | 23.24 | 48.15 | 44.77 | 23.24 | 47.44 | 21.47 | 16.77 | 59.33 | 20.70 | 14.17 | 56.03 |
| LGA | 58.29 | 24.80 | 30.33 | 35.07 | 30.51 | 57.85 | 31.65 | 28.44 | 60.56 | 16.92 | 14.16 | 63.88 | 10.02 | 17.86 | 66.71 |
| LANDER | 48.43 | 19.90 | 40.19 | - | - | - | - | - | - | - | - | - | - | - | - |
| Fed-L2P | 74.29 | 3.75 | 14.33 | 78.92 | 1.01 | 14.00 | 80.80 | 0.45 | 11.41 | 58.26 | 22.67 | 22.54 | 57.12 | 24.34 | 19.61 |
| Fed-DualP | 81.95 | -0.87 | 6.67 | 85.91 | -0.45 | 7.01 | 86.88 | 0.67 | 5.33 | 62.89 | 28.21 | 17.91 | 62.41 | 26.94 | 14.32 |
| Fed-CODAP | 68.72 | 21.62 | 19.90 | 80.11 | 15.12 | 12.81 | 80.13 | 15.70 | 12.08 | 37.55 | 42.26 | 43.25 | 39.80 | 45.68 | 36.93 |
| Fed-CPrompt | 73.82 | 24.69 | 14.80 | 88.77 | 8.29 | 4.15 | 86.34 | 12.38 | 5.87 | 61.23 | 37.33 | 19.57 | 58.26 | 36.86 | 18.47 |
| PILoRA | 51.78 | 26.88 | 36.84 | - | - | - | - | - | - | - | - | - | - | - | - |
| **UOPP (Ours)** | **88.62** | **5.83** | **0.00** | **92.92** | **0.73** | **0.00** | **92.21** | **2.18** | **0.00** | **80.80** | **10.90** | **0.00** | **76.73** | **19.54** | **0.00** |

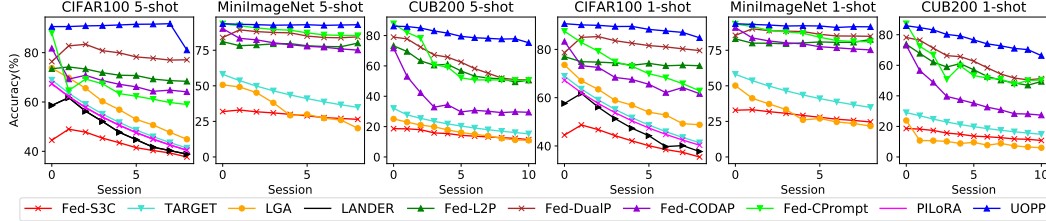

Figure 4: Visualization of the performance of consolidated algorithms in MiniImageNet, CIFAR100 and CUB200.

three datasets. In the first task (base task), the proposed method achieves higher performance with a small gap i.e. $1 - 2\%$ than the competitor methods. However, with the increasing number of tasks, the gap gets higher e.g. $\geq 3\%$ in task-1, $\geq 4\%$ in task-2, and $\geq 6\%$ in task-9 (last task). It shows that the proposed method handles catastrophic forgetting better than the competitor methods. Please see our extended analysis on different k-shot and standalone FSCIL in Appendix D. The complete numerical results for all dataset are presented in Appendix H.

**b) Stability-Plasticity Analysis:** We evaluate UOPP stability-plasticity performance by evaluating the harmonic mean accuracy on each few-shot task. Table 2 shows the harmonic mean accuracy of consolidated methods in CIFAR100 with 5-shot setting, while Figure 5 visualizes the accuracy for base classes, novel classes, and harmonic mean accuracy. Both show that UOPP achieves the best harmonic mean accuracy with $\geq 15\%$

Table 2: Harmonic mean accuracy of the consolidated algorithms in CIFAR100 dataset with 5-shot setting.

| Method | Harmonic Mean Acc. by Session (%) | | | | | | | | Avg | PD | Gap |
|---|---|---|---|---|---|---|---|---|---|---|---|
| | 1 | 2 | 3 | 4 | 5 | 6 | 7 | 8 | | | |
| Fed-S3C | 49.3 | 42.8 | 38.0 | 35.4 | 33.9 | 33.4 | 32.9 | 32.1 | 37.2 | 17.3 | 53.3 |
| TARGET | 0.0 | 0.0 | 0.0 | 0.0 | 0.0 | 0.0 | 0.0 | 0.0 | 0.0 | 0.0 | 90.5 |
| LGA | 44.7 | 31.5 | 19.3 | 16.2 | 12.5 | 12.3 | 9.9 | 7.8 | 19.3 | 36.9 | 71.2 |
| LANDER | 0.0 | 0.0 | 0.0 | 0.0 | 0.0 | 0.0 | 0.0 | 0.0 | 0.0 | 0.0 | 90.5 |
| Fed-L2P | 46.0 | 50.2 | 49.2 | 50.0 | 53.5 | 54.8 | 56.1 | 57.7 | 52.2 | -11.7 | 38.3 |
| Fed-DualP | 67.4 | 71.4 | 66.8 | 68.4 | 68.2 | 69.3 | 70.1 | 71.8 | 69.2 | -4.5 | 21.3 |
| Fed-CODAP | 69.8 | 69.3 | 66.1 | 65.2 | 63.3 | 61.8 | 62.0 | 61.3 | 64.9 | 8.5 | 25.6 |
| Fed-CPrompt | 73.3 | 74.4 | 67.6 | 63.5 | 62.3 | 60.4 | 59.4 | 58.9 | 65.0 | 14.3 | 25.5 |
| PILoRA | 0.0 | 0.0 | 0.0 | 0.0 | 0.0 | 0.0 | 0.0 | 0.0 | 0.0 | 0.0 | 90.5 |
| **UOPP** | **90.6** | **91.6** | **91.6** | **91.9** | **92.1** | **92.0** | **92.1** | **81.8** | **90.5** | **8.8** | **0.0** |

gap on each task and $18-58\%$ on average of all tasks. The results prove that UOPP handles stability-plasticity dilemmas better than its competitors. Looking at the base classes and novel classes performance, Figure 5 shows that UOPP achieves the highest accuracy of base classes and novel classes that are consistent through all tasks. Fed-L2P has increased base classes and novel classes accuracy, but the performance is still below UOPP with a significant gap. Fed-Dual, Fed-CODAP, and Fed-CPrompt have relatively stable base class accuracy but decreasing novel class accuracy. TARGET,

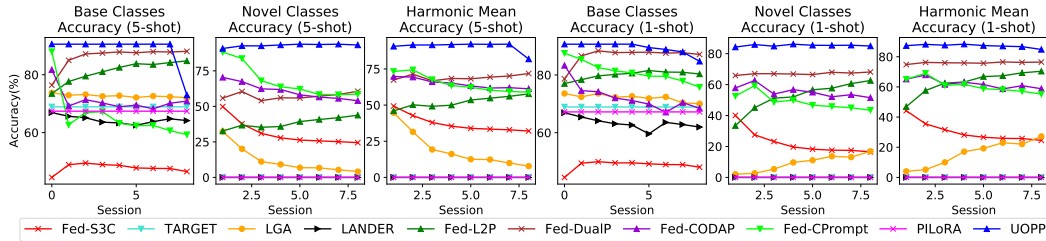

Figure 5: Visualization of the performance of consolidated methods for base classes and novel classes in CIFAR100 Dataset.

Table 3: Summary of the numerical result of the consolidated algorithms in CIFAR100 dataset with the variation of Non-i.i.d. level (a), selected local clients (b) and variation of total global round (c).

| Method | (a) Non-i.i.d level ($\eta$) | | | | | | (b) Selected Local Client (L) | | | | | | (c) Total Global Round (R) | | | | | |
|---|---|---|---|---|---|---|---|---|---|---|---|---|---|---|---|---|---|---|
| | $\eta=0.6$ (60%) | | $\eta=0.4$ (40%) | | $\eta=0.2$ (20%) | | L=4 (20%) | | L=6 (30%) | | L=8 (40%) | | R=54 (6 r/task) | | R=72 (8 r/task) | | R=90 (10 r/task) | |
| | Avg | PD | Avg | PD | Avg | PD | Avg | PD | Avg | PD | Avg | PD | Avg | PD | Avg | PD | Avg | PD |
| Fed-S3C | 43.2 | 6.8 | 34.2 | 0.5 | 18.9 | 1.4 | 43.9 | 4.2 | 43.2 | 6.8 | 42.7 | 6.2 | 43.4 | 5.3 | 47.4 | 10.1 | 42.6 | 6.5 |
| TARGET | 53.1 | 27.6 | 41.3 | 27.5 | 23.2 | 13.5 | 51.4 | 26.7 | 53.1 | 27.6 | 56.7 | 29.4 | 44.4 | 23.0 | 51.8 | 26.9 | 53.5 | 27.8 |
| LGA | 58.1 | 28.9 | 56.4 | 29.5 | 56.4 | 29.7 | 56.3 | 30.4 | 58.1 | 28.9 | 57.9 | 29.1 | 57.1 | 22.1 | 55.2 | 26.3 | 58.0 | 29.3 |
| LANDER | 49.1 | 19.7 | 37.6 | 18.9 | 17.5 | 8.9 | 47.4 | 22.3 | 49.1 | 19.7 | 51.6 | 20.8 | 37.9 | 31.4 | 47.2 | 26.0 | 49.1 | 19.7 |
| Fed-DualP | 79.3 | -0.7 | 60.2 | -41.9 | 6.6 | -21.5 | 76.2 | -11.2 | 79.3 | -0.7 | 79.2 | 11.1 | 81.5 | -0.9 | 80.2 | 1.8 | 79.3 | -0.7 |
| Fed-Cpompt | 66.1 | 29.0 | 66.2 | 28.1 | 66.2 | 28.1 | 62.5 | 36.9 | 66.1 | 29.0 | 59.6 | 38.1 | 62.5 | 36.9 | 59.6 | 38.1 | 66.1 | 29.0 |
| PILoRA | 51.9 | 27.0 | 43.4 | 22.5 | 35.3 | 18.3 | 50.6 | 26.3 | 51.9 | 27.0 | 51.1 | 26.5 | 55.9 | 29.0 | 53.9 | 28.0 | 51.9 | 27.0 |
| **UOPP** | **90.0** | **9.5** | **86.3** | **4.3** | **68.8** | **29.4** | **89.1** | **4.2** | **90.0** | **9.5** | **91.3** | **0.0** | **90.7** | **-1.5** | **90.8** | **0.2** | **90.0** | **9.5** |

LANDER, and PILoRA can't achieve plasticity since their novel classes' accuracy is close to 0. Fed-S3C maintains stability-plasticity dilemma with relative balances, but the performance of both components is low ($< 40\%$). The complete numerical results are presented in Appendix I.

## 6.3 ROBUSTNESS, ABLATION, AND FURTHER ANALYSIS:

**a) Different Non-i.i.d. level:** Table 3 (a) shows the performance of the methods w.r.t. non-i.i.d level represented by the percentage of available class $\eta$ (lower is harder). The table shows that our method outperforms the existing methods in all non-i.i.d. levels with a significant margin i.e. up to 50%. The table shows that the small available class ($\eta = 20\%$) remains challenging since all the methods achieve less than 70% on

Table 4: Summary of the numerical result of our ablation study in CIFAR100 dataset, MLP, and PB and Rect. denote prototypes-based and rectification respectively.

| Conf. | Stiatic Proto. | Dynamic Proto. | MLP Head | PB. Head | Rect. | Avg | PD | Gap | Time |
|---|---|---|---|---|---|---|---|---|---|
| A | - | ✓ | ✓ | ✓ | ✓ | 80.62 | 7.76 | 9.39 | 6.01h |
| B | ✓ | - | ✓ | ✓ | - | 85.32 | 10.06 | 4.69 | 4.17h |
| C | ✓ | ✓ | - | ✓ | ✓ | 88.72 | 5.49 | 1.29 | 8.00h |
| D | - | - | ✓ | - | - | 69.23 | 37.76 | 20.78 | 3.50h |
| UOPP | ✓ | ✓ | ✓ | ✓ | ✓ | 90.01 | 9.52 | 0.00 | 6.05h |

average. In contrast, in other cases i.e. ($\eta > 40\%$) our method achieves $> 86\%$ accuracy on average.

**b) Different Participating Local Clients:** we evaluate UOPP robustness in various selected local clients simulating fluctuations of participating clients in real-life applications. Table 3 (b) shows the performance of consolidated methods with 4 (20%), 6 (30%), and 8 (40%) selected local clients from 20 total clients. The table shows that the UOPP achieves the highest performance in all combinations with a significant gap i.e. $\geq 10\%$ compared to the competitor methods. Table 3 also shows that UOPP achieves higher performance on a higher percentage of participating local clients. Besides, UOPP achieves a relatively lower performance drop (PD) than Fed-S3C, TARGET, LGA, LANDER and Fed-CPrompt. In 8 (40%) local client cases, UOPP achieves the lowest PD, while in 4 (20%) and 6 (30%) local client cases, UOPP experiences a higher PD than 8 and 4 local client cases due to an accuracy drop on the last task. The accuracy drop is caused by a mismatch between the samples and prompt keys resulting in inaccurate feature extraction and classifiers selection. The more detailed per-task result is presented in Appendix J.

**c) Variation of Total Global Rounds:** we evaluate UOPP robustness in smaller global rounds simulating real-world conditions where the global model is urgently needed, thereby requiring smaller rounds. Table 3(b) summarizes our investigation on 54 (6 r/task) to 90 (10 r/task) global rounds in CIFAR100 datasets, while the complete result is presented in Appendix J. . The table shows that UOPP achieves the highest performance with a significant gap i.e. ($\geq 9\%$) compared to the competitors. In the lower global rounds, the UOPP achieves even better performance than in normal global rounds as it doesn't experience the accuracy drop aforementioned. UOPP also achieves the smallest

PD compared to the competitor methods in smaller rounds. Prompt-based method i.e. Fed-DualP is proven to be more robust than Fed-S3C, TARGET, LGA, LANDER, and Fed-CPrompt. Furthermore, those 3 methods achieve lower accuracy in the smaller global rounds. This finding confirms the robustness of our proposed method in the case of low global rounds.

**d) Ablation Study:** we conduct an ablation study to investigate the contribution of each component of the proposed method. The result is summarized in Table 4, while the detailed result is presented in Appendix K. The result shows that the absence of static prototype (Conf. A) and dynamic prototype (Conf. B) drops the average performance with $9.4\%$ and $4.7\%$ gap respectively. This result proves the importance of unified prototypes for prompt learning to deal with FFSCIL problems. The absence of the MLP head drops the performance with $1.3\%$ gap. This result shows the presence of MLP classifiers (Conf. C) contributes to the model prediction performance. Last, the absence of the prototype-based head (Conf. D) drops the performance with the most significant magnitude e.g. $20.8\%$. It proves that the PB classifier is a must to deal with few-shot tasks. Note that the presence of rectification (Rect.) follows the presence of dynamic prototype (e.g. in configuration B and D where dynamic prototype is absent, the rectification is absent in those configurations).

**e) The Importance of Prototype Rectification:** We analyze the impact of prototype rectification in our proposed method. Figure 6 shows the validation loss of novel classes with rectification and without rectification on CIFAR100 dataset with 5-shot and 1-shot settings. The figure shows that, without prototype rectification (red line), our method produces a far higher validation loss for novel classes as there are many misclassified samples. On the contrary, with prototype rectification (blue line), our method produces far smaller and more stable validation loss. The figure also shows that the difference in loss magnitude between the two variants is even higher in the 1-shot setting. This finding emphasizes the importance of prototype rectification in our method.

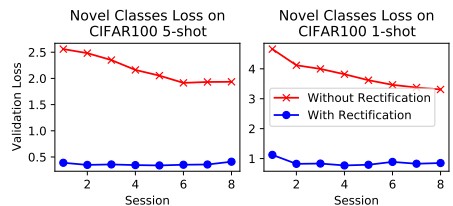

Figure 6: Visualization of Novel Classes Validation Loss with vs without Rectification.

**f) Complexity, Running Time, Parameters, and Communication Cost:** our complexity analysis shows that our proposed method has the same complexity i.e. $O(R.L.N)$ where $R$ is the number of global rounds, $L$ is the number of participating local clients in each round, and $N$ is the size of the training data in each client. The detailed complexity analysis is provided in Appendix C. Table 5 shows the training time of the consolidated method in CIFAR100 dataset. The table shows that the proposed method requires a moderate training time since it is lower than LGA and higher than the other methods.

Table 5: Comparison of Parameters, Communication Cost, and Training time in CIFAR100 dataset.

| Method | Number of Parameters (M) | | | | Comm. Cost (MB) | | Running Time (h) |
|---|---|---|---|---|---|---|---|
| | Trainable | | Sharable | | | | |
| | Base Task | FS Task | Base Task | FS Task | Base Task | FS Task | |
| Fed-S3C | 11.7 | 11.7 | 11.7 | 11.7 | 46.8 | 46.8 | 4.1 |
| TARGET | 11.3 | 11.3 | 17.45 | 17.45 | 61.7 | 61.7 | 2.02 |
| LGA | 11.3 | 11.3 | 15.42 | 15.42 | 69.8 | 69.8 | 8.22 |
| LANDER | 11.3 | 11.3 | 17.45 | 17.45 | 61.7 | 61.7 | 4.78 |
| Fed-L2P | 0.29 | 0.29 | 0.29 | 0.29 | 1.16 | 1.16 | 4.83 |
| Fed-DualP | 0.33 | 0.33 | 0.33 | 0.33 | 1.32 | 1.32 | 3.2 |
| Fed-CODAP | 0.33 | 0.33 | 0.33 | 0.33 | 1.32 | 1.32 | 1.42 |
| Fed-Cprompt | 0.33 | 0.33 | 0.33 | 0.33 | 1.32 | 1.32 | 2.03 |
| PILoRA | 0.30 | 0.30 | 0.34 | 0.37 | 1.37 | 1.50 | 6.24 |
| **UOPP** | **0.33** | **34.1** | **0.38** | **0.41** | **1.5** | **1.63** | **6.05** |

In the base task, UOPP trains only a small amount (0.33 M) of parameters, since it trains only its prompts and MLP classifier. However, in a few-shot tasks, it trains GradNet for prototype rectification which contributes to the high amount of trainable parameters. However, UOPP keeps its low communication cost both in the base task and few shot tasks since it only shares prompt+MLP or prompt+prototypes in both tasks. Please see Appendix D and E for our extended analysis on running time, memory consumption, limitations and potential solution.

## 7 CONCLUDING REMARKS

We define a new Federated Few-Shot Class-Incremental Learning (FFSCIL) problem and develop a novel Unified Optimized Prototype Prompt (UOPP) model that utilizes task-wise prompt learning to mitigate task interference empowered by shared static-dynamic prototypes, adaptive dual heads, and weighted aggregation. The dynamic prototype tackles prototype bias by iterative rectifications. Our comprehensive experimental results show that UOPP significantly outperforms existing SOTA methods of FCIL, FSCIL, and CIL, on three datasets with a significant gap i.e. up to $76\%$. Our deeper analysis confirms that the proposed method achieves better stability-plasticity trade-off, and robustness in different local clients and small global rounds. Our analysis shows that our proposed method requires moderate training time but a lower communication cost than the SOTAs.

ACKNOWLEDGMENTS

M. Anwar Ma'sum acknowledges the support of Tokopedia-UI Centre of Excellence for GPU access to run the experiments.

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

## A    DETAILED PROCESS OF UNIFIED OPTIMIZED PROTOTYPE PROMPT (UOPP)

In this section, we present the detailed algorithm of UOPP as shown in algorithm 1.

## B    DETAILED THEORETICAL ANALYSIS

Let $\Theta = (P, \Phi, \Psi)$ be trainable parameters, $F(\Theta) = \mathbb{E}[\mathcal{L}(\mathcal{T}; \Theta)] = \mathbb{E}[\mathcal{L}(\mathcal{T}; (P, \Phi, \Psi))]$ is the expected loss function, $k, E, R$, and $L_S$ is local iteration, local epoch, global round, and number of selected local clients respectively. We follow $L$-smooth and $\mu$-strongly convex $F$, $G$-bounded uniformly gradient assumptions, random uniformly distributed batches, and decreasing learning rate as in (Li et al., 2019),(Bottou et al., 2018) We state the following theorems.

**Theorem 1:** $\liminf_{k \to \infty} \mathbb{E}[||\nabla F(\Theta_k)||_2^2] = 0$

**Theorem 2:** Let $\Theta^1, \Theta^R, \Theta^*$ is the initial, last updated ($R$-th), and optimum parameter respectively, $F^*$ is minimum of $F$, exist $A, B, C, \delta > 0$ so that: $\mathbb{E}[F(\Theta^R)] - F^* \leq \frac{A}{R}(\frac{2(B+C)}{\mu} + \frac{\mu\delta}{2}\mathbb{E}||\Theta^1 - \Theta^*||)$.

**Theorem 3:** Given $\Theta^*$ and $\Theta$ are optimal parameter in $\mathcal{T}_l^t \cup Z$ and $\mathcal{T}_l^t$ respectively, where $\mathcal{T}_l^t \subset \mathcal{T}^t$, where $|\mathcal{T}_l^t|/|\mathcal{T}^t| = \eta \in (0, 1)$, then at least there's $\epsilon > 0$ that satisfy $F(\Theta; \mathcal{T}^t) - F(\Theta^*; \mathcal{T}^t) \geq \epsilon$. Theorem 1 and 2 prove UOPP local training and federated convergence respectively, while theorem 3 proves UOPP generalization. The detailed theoretical analysis, assumptions, and proofs are presented in Appendix B.

Let $\Theta = (P, \Phi, \Psi)$ be the trainable parameters, $F(\Theta) = \mathbb{E}[\mathcal{L}(\mathcal{T}; \Theta)] = \mathbb{E}[\mathcal{L}(\mathcal{T}; (P, \Phi, \Psi))]$ is the expected loss function, $k, E, R$, and $L$ is local iteration, local epoch, global round, and number of selected local clients respectively. Please note that in this analysis, $L$ denotes the number of selected local clients, while $l \geq 1$ denotes a constant for the $l$-smooth coefficient. Following the update rule in section 4.3, the expression of $F(\Theta)$ above can be detailed as follows:

**(i) Base Task** ($t = 0$): $\Theta = (P, \Phi)$, and $F(\Theta) = \mathbb{E}[\mathcal{L}(\mathcal{T}; \Theta)] = \mathbb{E}[\mathcal{L}_{l+}(\mathcal{T}; (P, \Phi))]$ as local clients update $(P, \Phi)$ using $\mathcal{L}_{l+}$ following equations 7 and 8.

**(i) FS Task** ($t \geq 1$): $\Theta = (P, \Psi)$, and $F(\Theta) = \mathbb{E}[\mathcal{L}(\mathcal{T}; \Theta)] = \mathbb{E}[\mathcal{L}_{lfs+}(\mathcal{T}; (P, \Psi))]$ as local clients update $(P, \Psi)$ using $\mathcal{L}_{lfs+}$ following equations 9 and 10.

We adopt the SGD optimization convergence analysis (Bottou et al., 2018) and FedAvg convergence analysis (Li et al., 2019) assumptions as follows:

---

**Algorithm 1** UOPP

---

1: **Input:** Number of clients $L_{all}$, number of selected local clients $L$, total number of rounds $R$, number of task $T + 1$, local epochs $E$, batch size $B$.
2: Distribute frozen ViT backbone $f$ to all clients $\{l\}_{l=1}^{L_{all}}$ and central server $G$
3: Initiate prompt, key, and head layer for all clients and central server $P_G = P_l$, $\Phi_G = \Phi_l$, $\Psi_l = init(), l \in \{1..L_{all}\}$
4: $R_T \leftarrow R/(T + 1)$, $R_T$ represents round per task
5: Init global and local unified prototypes $Z_G = Z_l, = Z = \emptyset$
6: **for** $t = 0 : T$ **do**
7:     **for** $r = 1 : R_T$ **do**
8:         $\{l\}_{l=1}^{L} \leftarrow$ randomly select $L$ local clients from $L_{all}$ total clients
9:         **Clients execute**:
10:         **if** $R_T = 1$ **then**
11:             Compute static prototype $\tilde{Z}_l$ as in Eq. (3)-(4), then send it to server
12:         **end if**
13:         Receive global parameters i.e. prompt, FC layer, and prototypes set $P_G, \Phi_G$, and $Z_G$
14:         Assign local parameters $(P_l, \Phi_l, Z_l) \leftarrow (P_G, \Phi_G, Z_G)$
15:         $\mathcal{B} \leftarrow$ Split $\mathcal{T}_l^t$ into $B$ sized batches
16:         **for** $e = 1 : E$ **do**
17:             **for** $b = 1 : \mathcal{B}$ **do**
18:                 **if** $(t = 0)$ **then** // Base Task Update
19:                     Compute prompt-generated feature $f_{(K_l^t, P_l^t)}(x)$ as in Eq. (2)
20:                     Compute logits with FC clsasifier $g_{\Phi_l}(f_{(K_l^t, P_l^t)}(x) \cup Z_G)$
21:                     Compute loss $\mathcal{L}_{t=0}$ as in Eq. (10)
22:                     Update local parameters $(K_l^t, P_l^t, \Phi_l)$ based on $\mathcal{L}_{t=0}$
23:                 **else** $(t > 0)$ // Few-shot Task Update
24:                     Compute static prototype $\tilde{Z}_l$ using feature $f_{(K_l^t, P_l^t)}(x)$ as in Eq. (2)
25:                     Draw $\mathcal{S}$ from $\tilde{Z}_l$ and draw $\mathcal{Q}$ from $Z_l = Z_G$
26:                     Rectify dynamic prototype $\hat{Z}_l$ using $g_{\Psi}(.)$ as in Eq. (5) to (7)
27:                     Form unified prototype $Z_l = Z_G \cup \hat{Z}_l$
28:                     Compute logits with PB classifier $g_{Z_l}(f_{(K_l^t, P_l^t)}(x) \cup \mathcal{S})$
29:                     Compute loss $\mathcal{L}_{t>0}$ as in Eq. (11)
30:                     Update local parameters $(K_l^t, P_l^t, \Psi_l)$ based on $\mathcal{L}_{t>0}$
31:                 **end if**
32:             **end for**
33:             **if** $t = 0$ **then**
34:                 Update local static prototype $\tilde{Z}_l$ as Eq. (3)-(4) for all class $c \in \mathcal{T}_l^t$
35:             **end if**
36:         **end for**
37:         **if** $t = 0$ **then**
38:             Set unified prototype $Z_l = \tilde{Z}_G \cup \tilde{Z}_l$
39:         **else**
40:             Set unified prototype $Z_l = \tilde{Z}_G \cup \hat{Z}_l$
41:         **end if**
42:         Store local parameters $(K_l^t, P_l^t, \Phi_l, \Psi_l, Z_l)$
43:         Compute clients' weight $\omega_l$
44:         Send local parameters $(K_l^t, P_l^t, \Phi_l, Z_l)$ and weight $\omega_l^t$ to server
45:         **Server executes**:
46:         **if** $R_T = 1$ **then**
47:             Receive clients initial static prototype $\tilde{Z}_l$ for $l \in [1..L]$
48:             Generate $Z_G = Z_G \cup Agg(\tilde{Z}_l$ for $l \in [1..L])$ and send $Z_G$ to clients
49:         **end if**
50:         Receives selected clients parameters $(K_l^t, P_l^t, \Phi_l, Z_l)$ and weight $\omega_l$ for $l \in [1..L]$
51:         Do weighted aggregation as in Eq. (9)
52:         Send global parameters $(K_G^t, P_G^t, \Phi_G, Z_G)$ to clients for the next round
53:     **end for**
54: **end for**
55: **Output:** Optimal Global parameters $(\mathbf{P}_G, \Phi_G, Z_G)$

---

**Assumption 1:** $F_1, ... F_l, ..., F_{L_S}$ are all $L-$smooth: for all $\Theta$ and $\Theta'$, $F_l(\Theta) \leq F_l(\Theta') + (\Theta - \Theta')^T \nabla F_l(\Theta) + \frac{L}{2}||\Theta - \Theta'||_2^2$.

**Assumption 2:** $F_1, ... F_l, ..., F_{L_S}$ are all $\mu-$strongly convex: for all $\Theta$ and $\Theta'$, $F_l(\Theta) \leq F_l(\Theta') + (\Theta - \Theta')^T \nabla F_l(\Theta) + \frac{\mu}{2}||\Theta - \Theta'||_2^2$.

**Assumption 3:** Let $\xi_l^k$ be the random uniformly sampled from $l$-th local data at $k - th$ iteration . The variance of stochastic gradients in each client is bounded by the following criteria: $\mathbb{E}||\nabla F_l(\Theta_l^k, \xi_l^k) - \nabla F_l(\Theta_l^k)|| \leq \sigma_l^2$ for $l = 1, 2, ..., L_S$

**Assumption 4:** The expected squared norm of stochastic gradients in each client is bounded by: $\mathbb{E}||\nabla F_l(\Theta_l^k, \xi_l^k)|| \leq G^2$ for all $l = 1, 2, ..., L_S$ and $k = 1, 2, ...., K$ where $K \in \mathbb{N}$.

**Assumption 5:** $\sum_{k=1}^{\infty} \alpha_l^k = \infty$ and $\sum_{k=1}^{\infty} \alpha_l^{k^2} < \infty$ where $\alpha_l^k$ is the learning rate of $l - th$ client in $k$-th step training.

We follow the theoretical analysis in federated class incremental learning method (Ma'sum et al., 2024) as it has a similar characteristic i.e. prompt-based method supported by shared prototypes.

### B.1    PROOF OF THEOREM 1

Let a client-$l$ be trained locally with its local data $\mathcal{T}_l^t \cup Z$, where $\mathcal{T}_l^t$ is local;y observed training samples for $t$-th task and $Z = Z_l = Z_G$ is aggregated unified prototype for task $t$ shared by server respectively. We assume that $Z$ is augmented so that $|z_{c_b}| \approx |x_{c_a}|$ for $z_{c_b} \in Z$ and $x_{c_a}^t \in \mathcal{T}_l^t \subseteq \mathcal{T}^t$. As an implication, the number of prototypes of unavailable classes in $\mathcal{T}_l^t$ and the samples of available classes in $\mathcal{T}_l^t$ are balanced. Then the local model $\Theta_l = (\mathbf{P}_l, \Phi_l)$ or $\Theta_l = (\mathbf{P}_l, \Psi_l)$ is updated in $K$ iterations based on minibatches drawn from $\mathcal{T}_l^t \cup Z$. Since thefeature extractor parameters are frozen, and $\mathcal{T}_l^t \cup Z$ has balance samples for all classes, then $\xi_l^k$ approximates $\xi^k$ that is a sample from $\mathcal{T}^t$. The local model parameters are optimized by a stochastic gradient (SG) approach. Suppose that $g(\Theta_l, \xi_l^k)$ is a SG function, then the parameter update can be simplified as:

$$\Theta_l^{k+1} \leftarrow \Theta_l^k - \alpha_l^k g(\Theta_l^k, \xi_l^k) \tag{A11}$$

Applying assumption 1, and local parameter updates $\Theta$ by iterating stochastic gradient with sample $\xi_l^k$, then we get:

$$F_l(\Theta_l^{k+1}) - F_l(\Theta_l^k) \leq (\Theta_l^{k+1} - \Theta_l^k)^T \nabla F_l(\Theta_l^k) + \frac{L}{2}||\Theta_l^{k+1} - \Theta_l^k||_2^2$$

$$\leq -\alpha_l^k \nabla F_l(\Theta_l^k)^T g(\Theta_l^k, \xi_l^k) + \alpha_l^{k^2} \frac{L}{2}||g(\Theta_l^k, \xi_l^k)||_2^2 \tag{A12}$$

The equation above can be derived into:

$$\mathbb{E}_{\xi_l^k}[F_l(\Theta_l^{k+1})] - F_l(\Theta_l^k) \leq -\alpha_l^k \nabla F_l(\Theta_l^k)^T \mathbb{E}[g(\Theta_l^k, \xi_l^k)]$$

$$+ \alpha_l^{k^2} \frac{L}{2} \mathbb{E}_{\xi_l^k}[||g(\Theta_l^k, \xi_l^k)||_2^2] \tag{A13}$$

The inequation above shows $\Theta_l^k$ optimization by SG method at a step $k$, and it shows the reduction of $F_l$ (left side) is bounded by the value in the right side involving $\nabla F_l$ which is derivative of $F_l$ at $\Theta_l^k$ along with $-g(\Theta_l^k, \xi_l^k)$ (first term) and second moment of $g(\Theta_l^k, \xi_l^k)$ (second term). Let $g(\Theta_l^k, \xi_l^k)$ be the unbiased estimator of $\nabla F_l$, then we derive inequation above into:

$$\mathbb{E}_{\xi_l^k}[F_l(\Theta_l^{k+1})] - F_l(\Theta_l^k) \leq -\alpha_l^k \nabla ||F_l(\Theta_l^k)||_2^2 + \alpha_l^{k^2} \frac{L}{2} \mathbb{E}_{\xi_l^k}[||g(\Theta_l^k, \xi_l^k)||_2^2] \tag{A14}$$

The inequation above guarantees SGD convergence as long as the stochastic directions and stepsize are chosen. We apply the restriction below to avoid the harm of the second term of the right side in the inequation above,

$$\mathbb{V}[g(\Theta_l^k, \xi_l^k)] = \mathbb{E}[||g(\Theta_l^k, \xi_l^k)||_2^2] - ||\mathbb{E}[g(\Theta_l^k, \xi_l^k)]||_2^2. \tag{A15}$$

Adopting first and second-moment limit as in (Bottou et al., 2018), then we add the following assumption.

**Assumption 6:** The objective function $F_l$ and SG satisfy the following conditions.

(a). The sequence of $\{\Theta_l^k\}$ is contained in an open space where $F_l$ is bounded below by a scalar $F_{inf}$

(b) Exist scalars $\nu_G \geq \nu > 0$ so that for all $k \in \mathbb{N}$ satisfy:

$$
\begin{aligned}
&\nabla F_l(\Theta_l^k)^T \mathbb{E}_{\xi_l^k}[g(\Theta_l^k, \xi_l^k)] \geq \nu||\nabla F_l(\Theta_l^k)T||_2^2, and \\
&||\mathbb{E}_{\xi_l^k}[g(\Theta_l^k, \xi_l^k)]||_2 \leq \nu_G||\nabla F_l(\Theta_l^k)||_2.
\end{aligned}
\tag{A16}
$$

(c) Exist scalars $m_1 \geq 0$ and $m_2 \geq 0$ so that for all $k \in \mathbb{N}$ satisfy:

$$
\mathbb{V}[g(\Theta_l^k, \xi_l^k)] \leq m_1 + m_2||\nabla F_l(\Theta_l^k)||_2^2
\tag{A17}
$$

Combining assumption 6 and restriction criteria as presented in equation (5), then we have:

$$
\begin{aligned}
\mathbb{E}_{\xi_l^k}[||g(\Theta_l^k, \xi_l^k)||_2^2] &\leq m_1 + m_G||\nabla F_l(\Theta_l^k)||_2^2, with \\
m_G &= m_2 + \nu_G^2 \geq \nu^2 > 0
\end{aligned}
\tag{A18}
$$

Then by substituting $\mathbb{E}_{\xi_l^k}[||g(\Theta_l^k, \xi_l^k)||_2^2]$ from equation (A8) into equation (A3), we have:

$$
\begin{aligned}
\mathbb{E}_{\xi_l^k}[F_l(\Theta_l^{k+1})] - F_l(\Theta_l^k) \leq & -\alpha_l^k \nabla F_l(\Theta_l^k)^T \mathbb{E}[g(\Theta_l^k, \xi_l^k)] \\
& + \alpha_l^{k^2} \frac{L}{2}(m_1 + m_G||\nabla F_l(\Theta_l^k)||_2^2)
\end{aligned}
\tag{A19}
$$

Assumption 5 ensures that $\{\alpha_l^k\} \to 0$ is practically achievable by applying a learning rate scheduler (with decay) that reduces the learning rate in each step of local training. Then by choosing $\alpha_l^k L m_G \leq \nu$ and substituting $\nabla F_l(\Theta_l^k)^T \mathbb{E}[g(\Theta_l^k, \xi_l^k)]$ in equation (A9) with the condition in assumption 6.b, we have

$$
\begin{aligned}
\mathbb{E}_{\xi_l^k}[F_l(\Theta_l^{k+1})] - F_l(\Theta_l^k) \leq & -\alpha_l^k \nu||\nabla F_l(\Theta_l^k)||_2^2 \\
& + \alpha_l^{k^2} \frac{L}{2}(m_1 + m_G||\nabla F_l(\Theta_l^k)||_2^2)
\end{aligned}
\tag{A20}
$$

Applying expectation into the equation above we get

$$
\begin{aligned}
\mathbb{E}_{\xi_l^k}[F_l(\Theta_l^{k+1})] - \mathbb{E}[F_l(\Theta_l^k)] \leq & -\alpha_l^k \nu \mathbb{E}[||\nabla F_l(\Theta_l^k)||_2^2] \\
& + \alpha_l^{k^2} \frac{1}{2}(m_1 + m_G \mathbb{E}[||\nabla F_l(\Theta_l^k)||_2^2]) \\
\mathbb{E}_{\xi_l^k}[F_l(\Theta_l^{k+1})] - \mathbb{E}[F_l(\Theta_l^k)] \leq & -\frac{1}{2}\nu\alpha_l^k \mathbb{E}[||\nabla F_l(\Theta_l^k)||_2^2] \\
& + \frac{1}{2}\alpha_l^{k^2} L m_1
\end{aligned}
\tag{A21}
$$

Sum both sides for $k \in \{1, ..., K\}$ we get

$$
\begin{aligned}
F_{inf} - \mathbb{E}[F(\Theta_l^1)] &\leq \mathbb{E}[F_l(\Theta_l^{K+1})] - \mathbb{E}[F_l(\Theta_l^1)] \\
F_{inf} - \mathbb{E}[F(\Theta_l^1)] &\leq -\frac{1}{2}\nu \sum_{k=1}^{K} \alpha_l^k \mathbb{E}[||\nabla F_l(\Theta_l^k)||_2^2] + \frac{1}{2}L m_1 \sum_{k=1}^{K} \alpha_l^{k^2}
\end{aligned}
\tag{A22}
$$

Dividing by $\nu$ for both sides, then we get

$$
\sum_{k=1}^{K} \alpha_l^k \mathbb{E}[||\nabla F_l(\Theta_l^k)||_2^2] \leq \frac{2(\mathbb{E}[F(\Theta_l^1)] - F_{inf})}{\nu} + \frac{L m_1}{\nu} \sum_{k=1}^{K} \alpha_l^{k^2}
\tag{A23}
$$

Applying $\lim_{K \to \infty}$ and assumption 5 to the equation above we get

$$
\begin{aligned}
\lim_{K \to \infty} \sum_{k=1}^{K} \alpha_l^k \mathbb{E}[||\nabla F_l(\Theta_l^k)||_2^2] \leq & \frac{2(\mathbb{E}[F(\Theta_l^1)] - F_{inf})}{\nu} \\
& + \frac{Lm_1}{\nu} \lim_{K \to \infty} \sum_{k=1}^{K} \alpha_l^{k^2} < \infty
\end{aligned} \tag{A24}
$$

Dividing both sides with $\sum_{k=1}^{K} \alpha_l^k$, and following assumption 5 where $\lim_{K \to \infty} \sum_{k=1}^{K} \alpha_l^k = \infty$ and $\lim_{K \to \infty} \sum_{k=1}^{K} \alpha_l^{k^2} < \infty$, then the right side will return 0. Therefore, we have

$$
\lim_{K \to \infty} \frac{\sum_{k=1}^{K} \mathbb{E}[\alpha_l^k ||\nabla F_l(\Theta_l^k)||_2^2]}{\sum_{k=1}^{K} \alpha_l^k} = 0 \tag{A25}
$$

$$
\lim_{K \to \infty} \mathbb{E}\left[\frac{\sum_{k=1}^{K} \alpha_l^k ||\nabla F_l(\Theta_l^k)||_2^2}{\sum_{k=1}^{K} \alpha_l^k}\right] = 0 \tag{A26}
$$

$$
\lim_{k \to \infty} \mathbb{E}[||\nabla F_l(\Theta_l^k)||_2^2] = 0 \tag{A27}
$$

The equation (A17) proves the convergence for local training in $l$-th client where the gradient of loss $F$ converges to 0 along with the increase of training step/iteration $k$ and the decreasing of learning rate $\alpha$.

## B.2 PROOF OF THEOREM 2

Let the selected local clients $\{l\}_{l=1}^{l=L_S}$ are conduct local optimization with its local training data $\{\mathcal{T}_l^t \cup Z\}_{l=1}^{l=L_S}$ coordinated by central server $G$, where $\mathcal{T}_l^t$ is local training sample for client $l$ for task $t$. Client local update is conducted in $k$ iterations using minibatch sampling on local training data set $\xi_l^k \in \mathcal{T}_l^t$. Global model aggregation is executed in each round $r = \{1, 2, ..., R\}$. We define global aggregation step as $\mathcal{I}_E = \{rE | r = 1, 2, ...R\}$. Following (Li et al., 2019), symbol $\Theta_l^{k+1}$ denotes the local parameter of client $l$ after communication steps, while $\varphi_l^{k+1}$ denotes the local parameter after an immediate result of one step of stochastic gradient descent. Then the definition satisfies the following expressions:

$$
\varphi_l^{k+1} = \Theta_l^k - \alpha_l^k \nabla F_l(\Theta_l^k, \xi_l^k) \tag{A28}
$$

$$
\Theta_l^{k+1} = \begin{cases} \varphi_l^{k+1} \text{ if } k+1 \notin \mathcal{I}_E \\ \sum_{l=1}^{L_S} w_l^k \varphi_l^{k+1} \text{ if } k+1 \in \mathcal{I}_E \end{cases} \tag{A29}
$$

Where $w_l = \omega_l / \sum_{l=1}^{L_S} \omega_l$, where $\omega_l$ is the weight of client $l$. We state $\bar{\varphi}_l^{k+1} = \sum_{l=1}^{L_S} w_l \varphi_l^{k+1}$ and $\bar{\Theta}_l^{k+1} = \sum_{l=1}^{L_S} w_l \Theta_l^{k+1}$, $\bar{\varphi}_l^{k+1}$ is the result of single step of stochastic gradient descent iteration from $\bar{\Theta}_l^{k+1}$. Then we define $\bar{g}^k = \sum_{l=1}^{L_S} w_l \nabla F_l(\Theta_l^k)$ and $g^k = \sum_{l=1}^{L_S} w_l \nabla F_l(\Theta_l^k, \xi_l^k)$. We adopt the lemmas below from (Li et al., 2019) where their derivation is obtained from fully participating clients in FL setting.

**Lemma 1:** By applying assumptions 1 and 2, in one step SGD training and chose $\alpha \leq \frac{1}{4L}$ we have

$\mathbb{E}[||\bar{\varphi}^{k+1} - \Theta^*||^2] \leq (1 - \alpha^k \mu)\mathbb{E}[||\bar{\Theta}^k - \Theta^*||^2] - (\alpha^k)^2 \mathbb{E}[||g^k - \bar{g}^k||^2] + 6L(\alpha^k)^2 \Gamma + 2\mathbb{E}[\sum_{l=1}^{L_S} w_l ||\bar{\Theta}^k - \Theta_l^k||^2]$ where $\Gamma = F^* - \sum_{l=1}^{L_S} w_l F_l^* \geq 0$.

**Lemma 2:** By applying assumption 3, the gradient function follows:

$\mathbb{E}[||\bar{g}^k - g^k||^2] \leq \sum_{l=1}^{L_S} w_l^2 \sigma_l^2$, where $\sigma_l^2$ is the variance of $\Theta_l$

**Lemma 3:** By applying assumption 4, where $\alpha^k$ is non-increasing and it satisfies $\alpha^k \leq \alpha^{k+E}$ for all $k \geq 0$, then we have $\mathbb{E}[\sum_{l=1}^{L_S} ||\bar{\Theta}^{k+1} - \Theta_l^k||^2] \leq 4(\alpha^k)^2(E-1)^2 G^2$

In FL setting with all clients participating, we get $\bar{\Theta}^{k+1} = \bar{\varphi}^{k+1}$. However, in a setting with only partial clients participating, we utilize a random sampling mechanism that satisfies $\mathbb{E}_{S_L}[\Theta^{k+1}] = \bar{\varphi}^{k+1}$. We also adopt the bounding condition from (Li et al., 2019) as shown in lemma 4.

**Lemma 4:** The expected difference between $\bar{\Theta}^{k+1}$ and $\bar{\varphi}^{k+1}$ bounded by : $\mathbb{E}_{S_L}[||\bar{\varphi}^{k+1} - \bar{\Theta}^{k+1}||^2] \leq \frac{4}{L_S}(\alpha^k)^2 E^2 G^2$ and in the case of $w_l$ is uniform for all $l$-th client, then $\mathbb{E}_{S_L}[||\bar{\varphi}^{k+1} - \bar{\Theta}^{k+1}||^2] \leq \frac{4(N_S - L_S)}{N_S - 1}(\alpha^k)^2 E^2 G^2$, where $N_S$ is total clients and $L_S$ is number of selected clients.

Please note that

$$||\bar{\Theta}^{k+1} - \Theta^*||^2 = ||\bar{\Theta}^{k+1} - \Theta^*||^2 \tag{A30}$$

$$||\bar{\Theta}^{k+1} - \Theta^*||^2 = ||\bar{\Theta}^{k+1} - \bar{\varphi}^{k+1} + -\bar{\varphi}^{k+1} - \Theta^*||^2 \tag{A31}$$

$$||\bar{\Theta}^{k+1} - \Theta^*||^2 = ||\bar{\Theta}^{k+1} - \bar{\varphi}^{k+1}||^2 + ||\bar{\varphi}^{k+1} - \Theta^*||^2 + 2||\bar{\Theta}^{k+1} - \bar{\varphi}^{k+1}||.||\bar{\varphi}^{k+1} - \Theta^*|| \tag{A32}$$

$$||\bar{\Theta}^{k+1} - \Theta^*||^2 = ||\bar{\Theta}^{k+1} - \bar{\varphi}^{k+1}||^2 + ||\bar{\varphi}^{k+1} - \Theta^*||^2 + 2\langle \bar{\Theta}^{k+1} - \bar{\varphi}^{k+1}, \bar{\varphi}^{k+1} - \Theta^* \rangle \tag{A33}$$

In the case of $k+1 \notin \mathcal{I}_E$, then the term $||\bar{\Theta}^{k+1} - \bar{\varphi}^{k+1}||^2$ vanishes. Then by applying lemma 4, we get

$$\mathbb{E}[||\bar{\Theta}^{k+1} - \Theta^*||^2] \leq (1 - \alpha^k \mu)\mathbb{E}[||\bar{\Theta}^{k+1} - \Theta^*||^2] + (\alpha^k)B \tag{A34}$$

In the case of $k+1 \in \mathcal{I}_E$, then by applying lemma 4, we get

$$\mathbb{E}[||\bar{\Theta}^{k+1} - \Theta^*||^2] \leq (1 - \alpha^k \mu)\mathbb{E}[||\bar{\Theta}^{k+1} - \Theta^*||^2] + (\alpha^k)(B + C) \tag{A35}$$

where $B = \sum_{l=1}^{L_S} w_l \sigma_l^2 + 6L\Sigma + 8(E-1)^2 G^2$ and $C = \frac{4(N_S - L_S)}{N_S - 1}(E^2 G^2)$ if $w_l$ is uniform and $C = \frac{4}{L_S}(E^2 G^2)$ otherwise.

By choosing $\alpha^k = \frac{\beta}{k + \delta}$ for some $\beta > 1/\mu$ and $\delta > 0$ so that $\alpha^1 \leq \min\{1/\mu, 1/4L\} = 1/4L$ and $\alpha^k \leq 2\alpha^{k+E}$ then we have $\mathbb{E}[||\bar{\Theta}^{k+1} - \Theta^*||^2] \leq \frac{v}{\delta + k}$ where $v = \max\{\frac{\beta^2(B+C)}{\beta\mu - 1}, (\delta + 1)||\bar{\Theta}^{k+1} - \Theta^*||^2\}$

Then, by applying a strong convexity assumption of $F$ we have

$$\mathbb{E}[\bar{\Theta}^k] - F^* \leq \frac{L}{2}\Delta^k \leq \frac{L}{2}\frac{v}{\delta + k} \tag{A36}$$

where $F^*$ is the minimum value of $F$ where optimum parameter $\Theta^*$ is achieved. Later on, if we choose $\beta = 2/\mu, \delta = \max\{8L/\mu, E\}$ and denote $\kappa = L/\mu, \alpha^k = 2/u(1/(\delta + k))$ then we have

$$\mathbb{E}[F(\bar{\Theta}^k)] - F^* \leq \frac{\kappa}{(\delta + k - 1)}\left(\frac{2(B + C)}{\mu} + \frac{\mu\delta}{2}\mathbb{E}||\Theta^1 - \Theta^*||\right) \tag{A37}$$

The equation above generalizes FL where the model is optimized in a total of $k$ iterations where in a practical implementation, $k = b.E.R, b$ is the number of batches. Then, we get $k > R$ as $E$ and $b$ are positive integers. Therefore, substituting $k$ with $R$ in the inequation above produces the higher amount on the right side. On that basis, the inequation above can be derived into:

$$\mathbb{E}[F(\Theta^R)] - F^* \leq \frac{\kappa}{(\delta + R - 1)}\left(\frac{2(B + C)}{\mu} + \frac{\mu\delta}{2}\mathbb{E}||\Theta^1 - \Theta^*||\right) \tag{A38}$$

The equation above can be derived into:

$$\mathbb{E}[F(\Theta^R)] - F^* \leq \frac{1}{R}\frac{\kappa}{(\delta/R + 1 - 1/R)}\left(\frac{2(B + C)}{\mu} + \frac{\mu\delta}{2}\mathbb{E}||\Theta^1 - \Theta^*||\right) \tag{A39}$$

Let $A = \frac{\kappa}{(\delta/R + 1 - 1/R)}$ is a positive number. Then the equation above can be derived into:

$$\mathbb{E}[F(\Theta^R)] - F^* \leq \frac{A}{R}\left(\frac{2(B + C)}{\mu} + \frac{\mu\delta}{2}\mathbb{E}||\Theta^1 - \Theta^*||\right) \tag{A40}$$

The inequation (A30) guarantees the proposed weighted federated learning achieves a convergence condition that is upper bounded by the amount on the right side.

### B.3 PROOF OF THEOREM 3

Let $\Theta^*$ and $\Theta$ are optimal parameter in $\mathcal{T}_l^t \cup Z$ and $\mathcal{T}_l^t$ respectively, where $\mathcal{T}_l^t \subset \mathcal{T}^t$, where $|\mathcal{T}_l^t|/|\mathcal{T}^t| = \eta \in (0, 1)$, then we have

$$F(\Theta; \mathcal{T}^t) = \eta F(\Theta; \mathcal{T}_l^t) + (1 - \eta)F(\Theta; (\mathcal{T}^t - \mathcal{T}_l^t)) \tag{A41}$$

$$F(\Theta^*; \mathcal{T}^t) = \eta F(\Theta^*; \mathcal{T}_l^t) + (1 - \eta)F(\Theta^*; (\mathcal{T}^t - \mathcal{T}_l^t)) \tag{A42}$$

Let that $\Theta^o$ is the initial value of $\Theta$ and $\Theta^*$ assigned by a random uniform initiation. Thus or all class $c \in \mathcal{T}_c^t = \mathcal{T}_{y=c}^t$ It satisfy $F(\Theta^o; \mathcal{T}_c^t) = e^o$. After optimally learning on $\mathcal{T}_l^t$ and $\mathcal{T}_l^t \cup Z$ then $\Theta^o$ become $\Theta$ and $\Theta^*$ respectively. Note that $\Theta$ knows only available classes in $\mathcal{T}_l^t$, while $\Theta^*$ knows classes that available both in $\mathcal{T}_l^t$ and classes in $\mathcal{T}^t - \mathcal{T}_l^t$ via $Z$. Assuming that the loss for predicting classes in $\mathcal{T}_l^t$ is $e^a < e^o$, then we have $F(\Theta; \mathcal{T}_l^t) = F(\Theta^*; \mathcal{T}_l^t) = e^a < e^o$. Since all the backbone parameters are frozen and $\Theta^*$ learn $Z$, then we get $F(\Theta; (\mathcal{T}^t - \mathcal{T}_l^t)) = e^o$, while $F(\Theta^*; (\mathcal{T}^t - \mathcal{T}_l^t)) = e^b$, where $e^a \geq e^b \geq e^o$.

Then, the equations (A41) and (42) can be derived to

$$F(\Theta; \mathcal{T}^t) = \eta e^a + (1 - \eta)e^o \tag{A43}$$

$$F(\Theta^*; \mathcal{T}^t) = \eta e^a + (1 - \eta)e^b \tag{A44}$$

Subtracting the equations above, then we have

$$F(\Theta; \mathcal{T}^t) - F(\Theta^*; \mathcal{T}^t) = \eta e^a + (1 - \eta)e^o - (\eta e^a + (1 - \eta)e^b) \tag{A45}$$

$$F(\Theta; \mathcal{T}^t) - F(\Theta^*; \mathcal{T}^t) = \eta e^a + (1 - \eta)e^o - \eta e^a - (1 - \eta)e^b \tag{A46}$$

$$F(\Theta; \mathcal{T}^t) - F(\Theta^*; \mathcal{T}^t) = (1 - \eta)e^o - (1 - \eta)e^b \tag{A47}$$

$$F(\Theta; \mathcal{T}^t) - F(\Theta^*; \mathcal{T}^t) = (1 - \eta)(e^o - e^b) \tag{A48}$$

As we have $0 < \eta < 1$ and $e^o > e^b$, the right side of the inequation above has a positive value. Then, by choosing a small positive number $\epsilon > 0$ where $(1 - \eta)(e^o - e^b) \geq \epsilon$ then we have.

$$F(\Theta; \mathcal{T}^t) - F(\Theta^*; \mathcal{T}^t) \geq \epsilon \tag{A49}$$

Inequation above proves that $\Theta^*$ is more generalized to $\mathcal{T}^t$ than $\Theta$. This shows that our idea i.e. empowering prompt learning with shared unified prototypes improves model generalization.

## C DETAILED COMPELXITY ANALYSIS

Following the pseudo-code in Algorithm 1, UOPP have several operations e.g. generate static prototype (line 11, 24, 34), drawing $\mathcal{S}, \mathcal{Q}$ from prototypes (line 25), Rectify prototype (line 26), updating model parameters (line 20-22, 28-30), forming unified prototype (line 27, 38, 40) data exchange between clients and server. Knowing that accumulating on all batches, generating prototype or compute features from $\mathcal{T}_l^t$ cost $O(N_l^t)$, drawing (augment) samples from feature costs $O(N_l^t)$, rectifying prototypes cost costs $O(N_l^t)$, parameters update cost costs $O(N_l^t)$, forming uniform prototype cost $O(1)$, and parameters exchange include aggregation costs $O(1)$, and we have 1 base task and T few-shot tasks (total task is (T+1)) then the UOPP complexity will be:

$$O(UPPP) = O(BaseTask) + O(FewStotTask) \tag{A50}$$

$$\begin{aligned} O(UOPP) =\, & O(1) + R_T(O(clients_{base}) + O(server_{base}) \\ & + O(1) + T.R_T.(O(clients_{fs}) + O(server_{fs})) \end{aligned} \tag{A51}$$

$$\begin{aligned} O(UOPP) =\, & O(1) + R_T.(L.O(1client_{base}) + O(server_{base})) \\ & + T.R_T.(L.O(1client_{fs}) + O(server_{fs})) \end{aligned} \tag{A52}$$

$$
\begin{aligned}
O(UOPP) =& O(1) + R_T.(L(O(N_l^0) + O(E.N_l^0) \\
& + O(E.N_l^0)) + O(1) \\
& + T.R_T.(L(O(N_l^t) + O(E.N_l^t) + O(E.N_l^t) \\
& + O(E.1) + O(E.N_l^t)) + O(1)
\end{aligned}
\tag{A53}
$$

$$
O(UOPP) = O(1) + R_T.L.O(E.N_l^0) + T.R_T.L.O(E.N_l^t)
\tag{A54}
$$

$$
O(UOPP) = O(1) + R_T.L.E.O(N_l^0) + T.R_T.L.E.O(N_l^t)
\tag{A55}
$$

$$
O(UOPP) = O(1) + R_T.L.E(O(N_l^0) + T.O(N_l^t))
\tag{A56}
$$

Please note that $N_l = N_l^0 + N_l^1 + ... + N_l^T = N_l^0 + T(N_l^t), t \in [1..T]$. Therefore, the equation above can be derived into:

$$
O(UOPP) = O(1) + R_T.L.E(O(N_l^0 + T.ON_l^t))
\tag{A57}
$$

$$
O(UOPP) = R_T.L.E.O(N_l)
\tag{A58}
$$

$$
O(UOPP) = O(R_T.L.E.N_l)
\tag{A59}
$$

Since E is set as a small constant in our method i.e. 1-20 and $R_T < R$, then the UOPP complexity will be:

$$
O(UOPP) = O(R.L.N_l)
\tag{A60}
$$

Our derivation shows that our proposed method has the complexity of $O(R.L.N_l)$ where $R$ is total global rounds, $L$ is the number of selected local clients in each round and $N_l$ is the number of samples in each client.

## D   EXTENDED EXPERIMENT RESULTS AND ANALYSIS

### D.1   EVALUATION ON STANDALONE FSCIL

We measure the performance of our proposed method in standalone FSCIL setting to evaluate our idea on prompting with a unified static-dynamic prototype and dual-head classifiers. We compare our method with existing SOTAs i.e. TEEN(NeurIPS, 2023)(Wang et al., 2024), NC-FSCIL(ICLR, 2023)(Yang et al., 2023), OrCo(CVPR, 2024)(Ahmed et al., 2024), and PriVi-Lege(CVPR, 2024)(Park et al., 2024). The evaluation is conducted in 3 datasets i.e. CIFAR100, MiniImageNet, and CUB200, following common settings in FSCIL.

Table A6 shows the detailed numerical result of our experiment. The table shows that our proposed method achieves a better performance in general. In comparison to TEEN, NC-FSCIL, and ORCO, our method archives a significantly better performance i.e. with more than $20\%$, $25\%$, and $14\%$ margin in CIFAR100, MiniImagenet, and CUB200 dataset respectively. In comparison to PriViLedge, Our method achieves higher performance in CIFAR100 and CUB200 dataset with $1.65\%$ and $4.9\%$ margins respectively. Our method achieves lower performance than PriViLedge in MiniImageNet dataset. Looking in a more derailed view, that lower performance is basically due to our method achieving lower performance in the base task i.e. $3\%$ lower accuracy. Thus, It affects the later (few-shot) tasks and average performance. Please note that PriviLedge utilize vision and language modality, while the rest methods only utilize vision modality. This factor should be considered into account in the performance comparison.

Looking from the performance drop (PD) metrics, our method archives the lowest performance drop in all dataset by a significant margin i.e. $2 - 28\%$. This indicates that our method handle the catastrophic forgetting better than the existing methods.

Table A6: Numerical result of the FSCIL methods in CIFAR100, MiniImageNet and CUB200 dataset with 5-shot setting, PD indicates the performance drop, and Gap indicates the gap between the respected method to our proposed method (UOPP).

| Method | Accuracy in each session (%) | | | | | | | | | | | Avg | PD | Gap |
|---|---|---|---|---|---|---|---|---|---|---|---|---|---|---|
| | **CIFAR100: 60 base classes @5 classes in few-shot task** | | | | | | | | | | | | | |
| | 0 | 1 | 2 | 3 | 4 | 5 | 6 | 7 | 8 | 9 | 10 | | | |
| TEEN (NeurIPS,2023) | 74.92 | 72.65 | 68.74 | 65.01 | 62.01 | 59.29 | 57.90 | 54.76 | 52.64 | - | - | 63.10 | 22.28 | 26.63 |
| NC-FSCIL (ICLR 2023) | 89.51 | 82.62 | 76.72 | 71.61 | 67.13 | 63.18 | 59.67 | 56.53 | 53.70 | - | - | 68.96 | 35.81 | 20.77 |
| OrCO (CVPR 2024) | 80.08 | 68.16 | 66.99 | 60.97 | 59.78 | 58.60 | 57.04 | 55.13 | 52.19 | - | - | 62.10 | 27.89 | 27.63 |
| PriViLege (CVPR 2024) | 90.88 | 89.39 | 88.97 | 87.55 | 87.83 | 87.35 | 87.53 | 87.15 | 86.06 | - | - | 88.08 | 4.82 | 1.65 |
| **UOPP (Ours)** | **90.27** | **90.45** | **90.14** | **89.40** | **89.26** | **89.52** | **89.70** | **89.48** | **89.35** | **-** | **-** | **89.73** | **0.92** | **0.00** |
| | **MiniImageNet: 60 base classes @5 classes in few-shot task** | | | | | | | | | | | | | |
| TEEN (NeurIPS,2023) | 73.53 | 70.55 | 66.37 | 63.23 | 60.53 | 57.95 | 55.24 | 53.44 | 52.08 | - | - | 61.44 | 21.45 | 31.58 |
| NC-FSCIL (ICLR 2023) | 77.25 | 71.30 | 66.21 | 61.80 | 57.94 | 54.53 | 51.50 | 48.79 | 46.35 | - | - | 59.52 | 30.90 | 33.50 |
| OrCO (CVPR 2024) | 83.30 | 75.32 | 71.53 | 68.16 | 65.63 | 63.12 | 60.20 | 58.82 | 58.08 | - | - | 67.13 | 25.22 | 25.89 |
| PriViLege (CVPR 2024) | 96.68 | 96.49 | 95.65 | 95.54 | 95.54 | 94.91 | 94.33 | 94.19 | 94.10 | - | - | 95.27 | 2.58 | -2.25 |
| **UOPP (Ours)** | **93.57** | **93.66** | **93.51** | **93.51** | **93.13** | **92.24** | **92.23** | **92.66** | **92.71** | **-** | **-** | **93.02** | **0.86** | **0.00** |
| | **CUB200: 100 base classes @10 classes in few-shot task** | | | | | | | | | | | | | |
| TEEN (NeurIPS,2023) | 77.26 | 76.13 | 72.81 | 68.16 | 67.77 | 64.40 | 63.25 | 62.29 | 61.19 | 60.32 | 59.31 | 68.14 | 17.95 | 14.82 |
| NC-FSCIL (ICLR 2023) | 78.49 | 71.52 | 65.54 | 60.30 | 55.81 | 51.96 | 48.72 | 45.78 | 43.18 | 40.92 | 38.80 | 57.92 | 39.69 | 25.04 |
| OrCO (CVPR 2024) | 75.59 | 66.85 | 64.05 | 63.69 | 62.20 | 60.38 | 60.18 | 59.20 | 58.00 | 58.42 | 57.94 | 63.35 | 17.66 | 19.61 |
| PriViLege (CVPR 2024) | 82.21 | 81.25 | 80.45 | 77.76 | 77.78 | 75.95 | 75.69 | 76.00 | 75.19 | 75.19 | 75.08 | 78.03 | 7.13 | 4.93 |
| **UOPP (Ours)** | **86.63** | **86.99** | **85.19** | **83.37** | **82.42** | **80.19** | **80.21** | **80.76** | **80.93** | **81.16** | **81.17** | **82.96** | **5.46** | **0.00** |

## D.2 Performance on Different k-Shot

We extend our investigation of the methods' performance on different k-shot values i.e. 7-shot, 5-shot, 3-shot, and 1-shot. Table A7 presents the detailed numerical results on CIFAR100 dataset with 1-7 shot settings. Table A7 shows that our proposed method consistently achieves the highest performance in different of k-shot values. The performance gap is consistently significant i.e. 7-46%. Regarding average performance, our method tends to achieve lower performance in lower k-shot values i.e. 1 and 3. UOPP performances in 5-shot and 7-setting are comparable on average. However, in terms of the performance in the last (final) session, UOPP achieves a better final performance in a 7-shot setting. This indicates that more samples for each client improve the global model performance.

This trend also applies to other methods where their performance on the higher shots is better than their performance on the lower shots. However, Fed-CPrompt shows the anomaly, where its performance in lower shots is better than its performance in higher shots. This condition indicates that the increase in novel classes' accuracy is lower than the drop in base classes' accuracy. Thus, is higher sample is contra-productive for few-shot task training. Looking to the performance drop (PD), the table shows that UOPP achieves a lower (better) performance drop in the higher k-shot value. This fact is in line with the mentioned trend of higher performance in the higher k-shot value before.

## D.3 Memory Consumption Analysis

We extended our resource analysis by evaluating the memory consumption by client and server. Table A8 shows the memory consumption by a single client and central server. Please note that each client trains its local model utilizing its local dataset, while the central server coordinates the clients, aggregates the locally optimized models into a global model, and redistributes it to the clients.

Table A8 shows the memory consumption for our proposed method, Fed-DualP, and PILoRA where these 3 methods have comparable communication costs and similar aggregation methods. As commonly known, Table A8 shows the memory consumption of deep learning training is highly affected by the batch size (BS) of the data loader, while the memory consumption for model aggregation is affected by the number of participating clients. Second, the table shows that the memory consumption for a central server is relatively lower than the client's memory consumption. Third, the memory consumption of our method is comparable with Fed-DualP, and PILoRa even though our method utilizes more operation for prototype rectification and has more trainable parameters during the few-shot tasks. This fact indicates that our method has comparable scalability and resource requirements with the existing methods. Last but not least, the table shows the high possibility of practical deployment of our method both in cross-silo and cross-device scenarios. The local training setting can be adjusted by the end device specification. FOr example, in the case of cross-device

Table A7: Numerical result of the FFSCIL methods in CIFAR100, 7-shot, 5-shot, 3-shot, and 1-shot setting, $S$ indicate number of shot, PD indicates the performance drop, and Gap indicates the gap between the respected method to our proposed method (UOPP).

| Method | S | Accuracy in each session (%) | | | | | | | | | Avg | PD | Gap |
|---|---|---|---|---|---|---|---|---|---|---|---|---|---|
| | | 0 | 1 | 2 | 3 | 4 | 5 | 6 | 7 | 8 | | | |
| Fed-S3C | 7 | 43.98 | 49.65 | 48.01 | 45.83 | 43.21 | 41.68 | 40.39 | 39.69 | 38.18 | 43.40 | 5.80 | 46.36 |
| TARGET | 7 | 68.65 | 63.40 | 58.36 | 52.68 | 50.04 | 46.58 | 43.96 | 39.37 | 34.04 | 50.78 | 34.61 | 38.98 |
| LGA | 7 | 73.15 | 67.08 | 64.56 | 59.87 | 56.15 | 52.04 | 50.00 | 47.29 | 44.39 | 57.17 | 28.76 | 32.59 |
| LANDER | 7 | 66.30 | 60.65 | 55.89 | 51.05 | 48.15 | 44.53 | 42.66 | 40.83 | 38.26 | 49.81 | 28.04 | 39.95 |
| Fed-DualP | 7 | 78.45 | 83.71 | 84.61 | 83.40 | 83.29 | 81.86 | 81.84 | 81.51 | 81.80 | 82.27 | -3.35 | 7.49 |
| Fed-Cprompt | 7 | 88.13 | 77.9385 | 69.6857 | 67.13 | 66.7125 | 64.8824 | 64.28 | 62.51 | 61.51 | 69.20 | 26.62 | 20.56 |
| PILoRA | 7 | 66.48 | 61.37 | 56.99 | 53.19 | 49.86 | 46.93 | 44.32 | 41.99 | 39.89 | 51.22 | 26.59 | 38.54 |
| **UOPP** | **7** | **90.77** | **90.86** | **91.04** | **90.75** | **91.08** | **89.35** | **88.84** | **87.87** | **87.32** | **89.76** | **3.45** | **0.00** |
| Fed-S3C | 5 | 44.51 | 48.97 | 47.77 | 45.35 | 43.48 | 41.47 | 40.33 | 39.32 | 37.71 | 43.21 | 6.80 | 46.80 |
| TARGET | 5 | 68.90 | 63.61 | 59.06 | 55.12 | 51.68 | 48.64 | 45.94 | 43.52 | 41.34 | 53.09 | 27.56 | 36.92 |
| LGA | 5 | 73.76 | 69.80 | 65.59 | 60.26 | 56.87 | 52.94 | 50.66 | 47.69 | 44.89 | 58.05 | 28.87 | 31.96 |
| LANDER | 5 | 58.60 | 61.75 | 56.26 | 52.11 | 47.71 | 44.71 | 41.69 | 40.28 | 38.87 | 49.11 | 19.73 | 40.90 |
| Fed-L2P | 5 | 73.47 | 74.20 | 73.37 | 71.88 | 70.85 | 70.72 | 69.28 | 68.66 | 68.37 | 71.20 | 5.10 | 18.81 |
| Fed-DualP | 5 | 76.39 | 82.75 | 83.37 | 80.80 | 79.93 | 78.26 | 77.73 | 76.98 | 77.11 | 79.26 | -0.72 | 10.75 |
| Fed-CODAP | 5 | 81.73 | 69.29 | 70.81 | 68.67 | 67.17 | 66.14 | 64.32 | 64.79 | 64.12 | 68.56 | 17.62 | 21.45 |
| Fed-Cprompt | 5 | 88.00 | 64.63 | 69.30 | 67.39 | 63.39 | 62.33 | 61.11 | 59.78 | 59.00 | 66.10 | 29.00 | 23.91 |
| PILoRA | 5 | 67.40 | 62.22 | 57.77 | 53.92 | 50.55 | 47.58 | 44.93 | 42.57 | 40.44 | 51.93 | 26.96 | 38.08 |
| **UOPP** | **5** | **90.57** | **90.58** | **90.85** | **90.96** | **91.23** | **91.51** | **91.56** | **91.74** | **81.05** | **90.01** | **9.52** | **0.00** |
| Fed-S3C | 3 | 43.98 | 49.60 | 48.03 | 45.19 | 42.69 | 41.12 | 39.87 | 38.82 | 37.47 | 42.97 | 6.51 | 45.64 |
| TARGET | 3 | 68.65 | 63.03 | 58.10 | 52.64 | 49.00 | 44.92 | 42.09 | 37.18 | 32.20 | 49.76 | 36.45 | 38.85 |
| LGA | 3 | 73.52 | 67.37 | 63.57 | 59.07 | 56.01 | 52.35 | 49.50 | 46.89 | 44.15 | 56.94 | 29.37 | 31.67 |
| LANDER | 3 | 66.30 | 59.77 | 55.56 | 50.56 | 47.40 | 44.04 | 41.72 | 40.72 | 38.43 | 49.39 | 27.87 | 39.22 |
| Fed-DualP | 3 | 73.20 | 79.31 | 82.73 | 81.53 | 82.06 | 81.16 | 81.56 | 81.21 | 80.34 | 80.34 | -7.14 | 8.27 |
| Fed-Cprompt | 3 | 88.32 | 83.62 | 81.46 | 77.68 | 77.18 | 75.40 | 74.32 | 70.63 | 68.51 | 77.46 | 19.81 | 11.15 |
| PILoRA | 3 | 66.43 | 61.32 | 56.94 | 53.15 | 49.83 | 46.89 | 44.29 | 41.96 | 39.86 | 51.19 | 26.57 | 37.42 |
| **UOPP** | **3** | **90.62** | **90.57** | **90.84** | **89.84** | **90.18** | **89.69** | **88.20** | **88.24** | **79.31** | **88.61** | **11.31** | **0.00** |
| Fed-S3C | 1 | 44.51 | 48.70 | 46.76 | 44.26 | 42.09 | 40.11 | 38.51 | 37.35 | 35.44 | 41.97 | 9.07 | 46.65 |
| TARGET | 1 | 68.90 | 63.61 | 59.06 | 55.12 | 51.68 | 48.64 | 45.94 | 43.52 | 41.34 | 53.09 | 27.56 | 35.53 |
| LGA | 1 | 73.58 | 67.00 | 63.44 | 58.88 | 56.90 | 54.00 | 52.82 | 49.25 | 48.78 | 58.29 | 24.80 | 30.33 |
| LANDER | 1 | 57.60 | 61.75 | 56.09 | 51.29 | 47.33 | 44.22 | 39.73 | 40.15 | 37.70 | 48.43 | 19.90 | 40.19 |
| Fed-L2P | 1 | 77.00 | 74.91 | 74.73 | 74.40 | 73.45 | 74.20 | 73.22 | 73.46 | 73.25 | 74.29 | 3.75 | 14.33 |
| Fed-DualP | 1 | 73.20 | 79.31 | 82.73 | 81.53 | 82.06 | 81.16 | 81.56 | 81.21 | 80.34 | 80.34 | -7.14 | 8.28 |
| Fed-CODAP | 1 | 83.29 | 73.27 | 72.54 | 68.37 | 67.52 | 65.54 | 62.11 | 64.19 | 61.67 | 68.72 | 21.62 | 19.90 |
| Fed-Cprompt | 1 | 87.53 | 82.86 | 79.24 | 74.87 | 73.04 | 69.92 | 68.28 | 65.81 | 62.84 | 73.82 | 24.69 | 14.80 |
| PILoRA | 1 | 67.20 | 62.03 | 57.60 | 53.76 | 50.40 | 47.44 | 44.80 | 42.44 | 40.32 | 51.78 | 26.88 | 36.84 |
| **UOPP** | **1** | **90.65** | **90.16** | **89.97** | **89.49** | **89.53** | **88.29** | **87.66** | **87.04** | **84.82** | **88.62** | **5.83** | **0.00** |

Table A8: Analysis of Memory Consumptions by Client and Server

| Method | Single Client Memory Consumption (GB) | | | | | |
|---|---|---|---|---|---|---|
| | BS=128 | BS=64 | BS=32 | BS=16 | BS=8 | BS=4 |
| Fed-DualP | 14.88 | 9.15 | 5.82 | 4.2 | 3.25 | 2.82 |
| PILoRA | 15.13 | 9.97 | 7.02 | 4.33 | 3.6 | 3.2 |
| UOPP | 14.89 | 8.68 | 5.59 | 3.98 | 3.26 | 2.83 |
| Method | Server Memory Consumption (GB) | | | | | |
| | L=1000 | L=100 | L=50 | L=10 | L=5 | L=2 |
| Fed-DualP | 1.32 | 0.1320 | 0.0660 | 0.0013 | 0.0066 | 0.0026 |
| PILoRA | 1.50 | 0.1495 | 0.0748 | 0.0015 | 0.0075 | 0.0030 |
| UOPP | 1.63 | 0.1627 | 0.0814 | 0.0016 | 0.0081 | 0.0033 |

Table A9: Running Time w.r.t. Percentage of Available Classes Number of Selected Local Client in Each Round

| Method | Run time (h) in different available classes percentage ($\eta$) | | | Run time (h) in different selected local clients (L) | | |
|---|---|---|---|---|---|---|
| | $\eta$=0.6(60%) | $\eta$=0.4(40%) | $\eta$=0.2(20%) | L=4 | L=6 | L=8 |
| Fed-S3C | 4.10 | 2.07 | 2.08 | 2.18 | 4.10 | 6.23 |
| TARGET | 2.02 | 2.15 | 1.99 | 1.50 | 2.02 | 2.50 |
| LGA | 8.22 | 5.35 | 5.03 | 3.51 | 8.22 | 15.23 |
| LANDER | 4.78 | 1.90 | 1.85 | 3.90 | 4.78 | 4.60 |
| Fed-DualP | 3.20 | 2.17 | 1.90 | 2.07 | 3.20 | 4.07 |
| Fed-Cpompt | 2.03 | 2.37 | 2.45 | 1.50 | 2.03 | 2.77 |
| PILoRA | 6.17 | 4.78 | 3.13 | 4.12 | 6.17 | 8.62 |
| UOPP | 6.05 | 5.87 | 5.35 | 4.70 | 6.05 | 8.81 |

settings where local training is conducted by small edge devices such as laptops or IoT nodes, the batch size can be set to a smaller value i.e. 16 or less. In the case of a cross-silo setting, where local training is conducted by more powerful end devices such as a corporate server, the batch size can be chosen by the larger value i.e. 128 or more.

## D.4 RUNNING TIME ANALYSIS

We extend our scalability analysis by investigating the simulation time w.r.t. the training data size e.g. indicated by the available classes percentage ($\eta$), and the number of participating local clients (L). table A9 shows the running time of our simulation w.r.t. those 2 factors. Please note that our simulation is run on a single GPU device. Thus each local training is executed sequentially. Thus, If the local training is conducted in a parallel way, the difference in the simulation time will be smaller.

The table shows that the ratio of the training data doesn't imply the same running time ratio. For example in the case of $\eta = 0.6$ where each client carries 3x the number of samples of $\eta = 0.2$, the running time in those $\eta = 0.6$ is not equal to 3x training time of $\eta = 0.2$. In our method, the increase of running time in those case is less than 20% (5.35 to 6.05). Thus, we found that the scale-up ratio of training samples with $r$ factor will impact far less than $r$ training time. Second, as we mentioned earlier regarding the sequential process of local training, table A9 shows a linear trend which is a logical result. For example, for UOPP and LORA, the total simulation time can be formulated by L x 1h. In the real application, local training should be conducted parallelly since each client runs on its local device. The bottleneck will be shifted to the server that receives the locally trained models, and the delay for round-trip communication (model transfer) between clients and server.

## E DISCUSSION ON LIMITATION AND POTENTIAL SOLUTION

Our study has several limitations that can be improved in future studies.

**a) Same $\eta$ for each client :** First, In our simulation, each client has the same non-i.i.d level represented by the same percentage of available classes ($\eta$). In the real application, each client may have a different degree of class availability from the other client. Thus, in the future study, the simulation can be extended into a variation percentage of available classes where each client is assigned with a random (picked from a min-max) $\eta$ range. This variable $\eta$ raises a new challenge for FFSCIL aside from simulating a more realistic setting.

**b) Simulation on a single device (GPU) :** Second, Our simulation is conducted in a single GPU, where each local training is executed sequentially one by one. This limitation will produce a linearly

increasing training time with the number of selected local clients. It is less realistic especially when we want to measure the training time. This limitation can be solved by utilizing a server/workstation that has multiple GPUs such as a Nvidia DGX server. The other solution is utilizing multiple cloud devices/servers as the clients.

**c) Fixed-size of Prompt :** In this study, our method utilized a fixed-size prompt for all clients. Related to the randomly selected available class and the new challenge of different $\eta$ (point a), It will be more realistic if a client decides its prompt size following the condition of its local data. The evolving prompt approaches such as ConvPrompt(Roy et al., 2024) and EvoPrompt(Kurniawan et al., 2024) may be suitable for those case. However, It raises a new challenge in the aggregation process i.e. how to produce an optimum global model from the different-sized local models which optimum for the current task as well as previously learned tasks.

**d) Overfitting Handling :** The current version of our method doesn't utilize advanced overfitting handling. Thus, in some cases e.g. in the CIFAR100 dataset, the model may suffer from overfitting indicated by the performance drop in the last task. One of the potential solutions is by applying early stopping during the training process. The other potential solution is applying learning decay to reduce the learning rate in line with the increasing training epochs.

**e) Multi-Modality :** The current version of our proposed method utilizes vision modality only. In the advance of Vision-Language Models, a language-guided approach may become a prospective approach to improve the model performance. Alternatively, language embedding can be utilized for prototype rectification instead of the prototypes generated by ViT.

## F  EXPERIMENT DETAILS AND HYPER PARAMETER SETTING

**Experimental Details:** our numerical study is executed under a single NVIDIA A100 GPU with 40 GB memory across 3 runs with different random seeds. Fed-L2P, Fed-DualP, Fed-CODAP, and UOPP train $T$ number of prompts $P \in R^{5 \times 768}$ and $\Phi \in R^{|C \times 768|}$ and head layer $\Phi$, while the competitors train whole CNN models following their original implementation. Following (Dong et al., 2023), each experiment is simulated by 20 total clients and 1 global server, where in each round, 6 (30%) local clients are selected randomly. Each client randomly receives 60% ($\eta = 0.6$) classes. The total global round is set to 90 (10 rounds per class) for CIFAR100 and MiniImageNet and 110 for CUB200. For all methods, the local training on each client is set with a maximum of 20 epochs, and the learning rate is set by choosing the best value from $\{0.001, 5.0\}$ by grid search with 2 incremental factors. Our setting is different to the recent study (Jiang et al., 2024), since it follows FCIL setting, while our setting follows FSCIL setting for the number of tasks, base classes, and novel classes in the few-shot tasks.

**Performance Metric:** on each session, we evaluate the consolidated algorithms to all learned classes with accuracy metrics (Acc(.)). Besides, we also measure the accuracy of base classes, novel classes and harmonic mean accuracy that indicates the balance between the performance of base classes and novel classes, in other words, it represents stability-plasticity performance. We also measure performance drop (PD), the accuracy difference between the first task, and the last task.

**CNN-Based Methods:** The competitor methods i.e. Fed-S3C, LGA, TARGET, and LANDER run with 2-20 local epochs on each client. The learning rate is set with the best result from 0.001 to 5.0 by 5 or 10 increment factor. The other hyperparameters such as weight decay, momentum, and dropout rate are set with their original setting. The methods utilize ResNet18 as the backbone model. LGA utilizes LeNet as the perturbation model. TARGET and LANDER use CNN as their synthesizer model. TARGET and LANDER generate 10000-50000 synthetic images on each task.

**Prompt-Based Methods:** PILoRA and the prompt-based methods i.e. Fed-L2P, Fed-DualP, Fed-CODAP, Fed-CPrompt, and UOPP, are run with ViT backbone. The base task is run with 1-2 epochs, while the few-shot task is run with 2-20 local epochs. For UOPP, the rectification step M is set to 40 steps per iteration. The initial learning rate is set with the best result from 0.001 to 0.2 by a 2 or 5 increment factor. The learning rate for the FS task and base task may be different. The prompt length is set to 5. The dual-head selection is executed in a batch-wise manner for convenience in implementation. The other parameters are set with the default settings.

## G    EXTENDED LITERATURE STUDY

**Few Shot Class Incremental Learning (FSCIL):** Previous studies on FSCIL have attempted to maintain stability-plasticity tradeoff in few labeled sequences of tasks by adding extra representation e.g. TOPIC (Tao et al., 2020b) introduces Neural gas as the graph of mapped features and CEC (Zhang et al., 2021) continually evolves its classifier to adapt to new tasks. Another approach modifies its learning mechanism e.g. FSLL (Mazumder et al., 2021) takes a partial parameter of the model to be updated with self-supervised loss, F2M (Shi et al., 2021) finds flat minima regions on the base task then forces parameter update on few shot tasks to reside within the flat region, S3C (Kalla & Biswas, 2022) trains scholastic classifier with supervised loss and MgSvF (Zhao et al., 2024) applies multi grained fast-slow learning mechanism. FSCIL methods demonstrate that representation or prototypes-based inference tends to be more stable (less forgetting) than linear classifiers under the data scarcity constraint. Nevertheless, the prototypes have to be still refined to avoid the prototype bias problem due to the data scarcity issue.

**Class Incremental Learning (CIL):** L2P (Wang et al., 2022b), DualP (Wang et al., 2022a), CODA-P (Smith et al., 2023) offer a breakthrough solution for CIL by training small-sized task-wise parameters called **prompts** while the feature extractor e.g. ViT that contains the biggest parameter numbers stays frozen. It solves task interference because each task has a specific prompt parameter to train based on a trainable matching key with its sample. This approach simplifies the training process and reduces memory consumption. The prompt-based approach is proven to be more effective than the rehearsal approach e.g. ICARL (Rebuffi et al., 2017), EEIL, (Castro et al., 2018), GD (Prabhu et al., 2020), DER++ (Buzzega et al., 2020), that saves exemplars from the previous tasks and replays them along with current task samples, the bias correction approach e.g. BiC (Wu et al., 2019) and LUCIR (Hou et al., 2019) that trains an additional task-wise bias layer to balance the model's stability-plasticity dilemma, and the regularization approach e.g. EWC (Kirkpatrick et al., 2017), MAS (Aljundi et al., 2018), LWF (Li & Hoiem, 2017), and DMC (Zhang et al., 2020) that tunes the base learner parameters to accommodate the previous task and current task. Regardless of its excellent performance in CIL, the prompt-based approach has not yet been proven in federated or few sample settings.

**Few Shot Learning (FSL):** FSL method e.g. metric learning(Ge, 2018), prototype network (Laenen & Bertinetto, 2021), and Neural ODE (Chen et al., 2018; Zhang et al., 2022) works effectively with few labeled training samples in a single session but not yet tested in continual or federated setting. However, it confirms that optimizing the prototypes tackles prototype bias that improves model performance greatly.

**Prototype-based Methods:** Prototype-based FSCIL method such as TEEN (Wang et al., 2024), NC-FSCIL (Yang et al., 2023), and OrCO(Ahmed et al., 2024) shows an insight the importance of prototype adjustment. TEEN recalibrates the prototypes using a similarity ratio between a calibrated prototype to base classes prototype and a novel classes prototype. NC-FSCIL utilizes the neural collapse principle for prototype alignment, while OrCo generates multi-angle prototypes to improve class representation and discrimination. Proroype-based FL methods such as FedPCL(Tan et al., 2022) and FedNCM(Legate et al., 2024) show how to conduct a prototype learning in a federated way, where a set of clients coordinated by a central server work together to achieve globally optimal prototypes. Prototype-based FCIL methods such as PILoRA show how a learnable prototype improves a parameter-efficient fine-tuning method to handle catastrophic forgetting with data-privacy constrains.

The strengths and weaknesses of CIL, FSL, FSCIL, and FCIL methods above inspire us to tackle the FFSCIL problem by developing a rehearsal-free prompt learning method combined with optimal prototypes to minimize communication costs.

## H    DETAILED NUMERICAL RESULTS ON BENCHMARK DATASETS

In this section, we present the detailed numerical result as shown in Tables A10, and A11,.

Table A10: Numerical result of the consolidated algorithms in MiniImageNet dataset with 5-shot and 1-shot setting across 3 different seeded runs. $S$ indicates the number of shots for the few shot tasks, PD indicates the performance drop, and Gap indicates the gap between the respected method to our proposed method (UOPP).

| Method | S | Accuracy in each session (%) | | | | | | | | | Avg | PD | Gap |
|---|---|---|---|---|---|---|---|---|---|---|---|---|---|
| | | 0 | 1 | 2 | 3 | 4 | 5 | 6 | 7 | 8 | | | |
| Fed-S3C | 5 | 31.91 | 32.97 | 31.74 | 30.96 | 29.93 | 28.92 | 27.66 | 27.06 | 26.43 | 29.73 | 5.5 | 63.2 |
| TARGET | 5 | 58.10 | 53.64 | 49.80 | 46.48 | 43.58 | 41.02 | 38.74 | 36.70 | 34.86 | 44.77 | 23.2 | 48.2 |
| LGA | 5 | 50.68 | 49.20 | 45.19 | 38.05 | 29.24 | 29.91 | 27.26 | 25.94 | 20.17 | 35.07 | 30.5 | 57.8 |
| Fed-L2P | 5 | 81.02 | 78.22 | 78.66 | 79.44 | 79.67 | 78.14 | 77.65 | 77.48 | 80.00 | 78.92 | 1.0 | 14.0 |
| Fed-DualP | 5 | 83.93 | 89.31 | 88.11 | 87.47 | 87.23 | 84.97 | 83.96 | 83.78 | 84.38 | 85.91 | -0.4 | 7.0 |
| Fed-CODAP | 5 | 90.21 | 83.35 | 82.31 | 80.27 | 78.89 | 77.79 | 77.13 | 75.94 | 75.09 | 80.11 | 15.1 | 12.8 |
| Fed-Cprompt | 5 | 93.57 | 92.26 | 90.71 | 89.60 | 89.09 | 87.22 | 85.80 | 85.41 | 85.28 | 88.77 | 8.29 | 4.15 |
| UOPP | 5 | 93.65 | 93.24 | 92.97 | 92.60 | 92.73 | 92.92 | 92.49 | 92.73 | 92.92 | 92.92 | 0.7 | 0.0 |
| Fed-S3C | 1 | 32.84 | 33.19 | 31.83 | 30.69 | 29.17 | 27.90 | 26.54 | 25.68 | 24.60 | 29.16 | 8.2 | 63.8 |
| TARGET | 1 | 58.10 | 53.64 | 49.80 | 46.48 | 43.58 | 41.02 | 38.74 | 36.70 | 34.86 | 44.77 | 23.2 | 48.2 |
| LGA | 1 | 50.15 | 41.26 | 37.24 | 33.33 | 26.04 | 27.02 | 24.61 | 23.53 | 21.71 | 31.65 | 28.44 | 60.56 |
| Fed-L2P | 1 | 83.02 | 79.99 | 79.92 | 79.54 | 80.20 | 80.84 | 80.55 | 80.60 | 82.57 | 80.80 | 0.4 | 12.1 |
| Fed-DualP | 1 | 85.47 | 90.06 | 88.77 | 88.44 | 88.14 | 86.22 | 85.04 | 84.98 | 84.80 | 86.88 | 0.7 | 6.0 |
| Fed-CODAP | 1 | 90.94 | 83.65 | 81.53 | 80.21 | 79.50 | 77.48 | 76.71 | 75.89 | 75.24 | 80.13 | 15.7 | 12.8 |
| Fed-Cprompt | 1 | 93.42 | 91.72 | 88.94 | 87.89 | 87.00 | 84.12 | 81.82 | 81.15 | 81.04 | 86.34 | 12.38 | 5.87 |
| UOPP | 1 | 93.66 | 93.15 | 92.72 | 92.04 | 92.20 | 92.05 | 91.03 | 91.56 | 91.48 | 92.21 | 2.2 | 0.0 |

Table A11: Numerical result of the consolidated algorithms in CUB200 dataset with 5-shot and 1-sot setting across 3 different seeded runs. $S$ indicates the number of shots for the few shot tasks, PD indicates the performance drop, and Gap indicates the gap between the respected method to our proposed method (UOPP).

| Method | S | Accuracy in each session (%) | | | | | | | | | | | Avg | PD | Gap |
|---|---|---|---|---|---|---|---|---|---|---|---|---|---|---|---|
| | | 0 | 1 | 2 | 3 | 4 | 5 | 6 | 7 | 8 | 9 | 10 | | | |
| Fed-S3C | 5 | 18.65 | 18.54 | 17.97 | 15.85 | 15.41 | 14.26 | 13.70 | 13.19 | 12.70 | 12.27 | 11.43 | 14.91 | 7.22 | 65.89 |
| TARGET | 5 | 32.03 | 27.75 | 25.44 | 23.48 | 21.80 | 20.35 | 19.08 | 17.96 | 16.96 | 16.07 | 15.26 | 21.47 | 16.77 | 59.33 |
| LGA | 5 | 25.07 | 22.95 | 21.03 | 19.89 | 17.86 | 16.11 | 14.94 | 13.88 | 12.26 | 11.19 | 11.09 | 16.92 | 14.16 | 63.88 |
| Fed-L2P | 5 | 73.24 | 69.49 | 63.32 | 60.46 | 61.05 | 56.65 | 53.24 | 51.98 | 51.47 | 49.42 | 50.57 | 58.26 | 22.67 | 22.54 |
| Fed-DualP | 5 | 78.98 | 77.83 | 72.34 | 67.29 | 65.75 | 62.44 | 58.25 | 55.17 | 51.99 | 50.95 | 50.77 | 62.89 | 28.21 | 17.91 |
| Fed-CODAP | 5 | 71.69 | 53.03 | 42.26 | 32.81 | 34.38 | 29.69 | 30.73 | 30.10 | 29.24 | 29.73 | 29.44 | 37.55 | 42.26 | 43.25 |
| Fed-CPrompt | 5 | 87.81 | 82.02 | 78.28 | 60.76 | 59.24 | 52.15 | 50.76 | 50.78 | 50.77 | 50.53 | 50.48 | 61.23 | 37.33 | 19.57 |
| UOPP | 5 | 86.18 | 85.95 | 84.96 | 83.02 | 81.62 | 79.48 | 78.57 | 78.15 | 77.70 | 77.86 | 75.28 | 80.80 | 10.90 | 0.00 |
| Fed-S3C | 1 | 18.65 | 18.22 | 17.34 | 15.65 | 14.86 | 13.64 | 13.22 | 12.64 | 11.80 | 11.57 | 10.79 | 14.40 | 7.85 | 62.33 |
| TARGET | 1 | 29.03 | 27.01 | 24.76 | 22.86 | 21.22 | 19.81 | 18.57 | 17.48 | 16.51 | 15.64 | 14.86 | 20.70 | 14.17 | 56.03 |
| LGA | 1 | 23.87 | 10.74 | 10.67 | 10.17 | 8.83 | 9.55 | 7.71 | 8.88 | 7.25 | 6.55 | 6.01 | 10.02 | 17.86 | 66.71 |
| Fed-L2P | 1 | 73.74 | 67.78 | 62.05 | 58.91 | 61.08 | 57.03 | 52.70 | 50.25 | 48.41 | 46.96 | 49.40 | 57.12 | 24.34 | 19.61 |
| Fed-DualP | 1 | 78.20 | 76.41 | 71.23 | 66.34 | 65.63 | 62.69 | 58.37 | 54.25 | 51.67 | 50.48 | 51.27 | 62.41 | 26.94 | 14.32 |
| Fed-CODAP | 1 | 73.07 | 56.54 | 48.81 | 39.62 | 37.46 | 35.15 | 32.56 | 30.81 | 28.32 | 28.10 | 27.39 | 39.80 | 45.68 | 36.93 |
| Fed-CPrompt | 1 | 87.22 | 72.41 | 66.88 | 50.86 | 59.63 | 53.31 | 51.58 | 50.99 | 47.60 | 50.02 | 50.36 | 58.26 | 36.86 | 18.47 |
| UOPP | 1 | 85.88 | 84.66 | 83.17 | 80.19 | 79.19 | 76.57 | 74.02 | 72.71 | 71.05 | 70.26 | 66.34 | 76.73 | 19.54 | 0.00 |

Table A12: Base classes accuracy of the consolidated algorithms in CIFAR100 dataset with 5-shot and 1-shot setting across 3 different seeded runs. $S$ indicates the number of shots for the few shot tasks, PD indicates the performance drop, and Gap indicates the gap between the respected method to our proposed method (UOPP).

| Method | S | Base Classes Accuracy in each session (%) | | | | | | | | | Avg | PD | Gap |
|---|---|---|---|---|---|---|---|---|---|---|---|---|---|
| | | 0 | 1 | 2 | 3 | 4 | 5 | 6 | 7 | 8 | | | |
| Fed-S3C | 5 | 44.51 | 48.90 | 49.46 | 48.93 | 48.69 | 47.78 | 47.64 | 47.58 | 46.54 | 47.78 | -3.07 | 40.83 |
| TARGET | 5 | 68.91 | 68.91 | 68.91 | 68.91 | 68.91 | 68.91 | 68.91 | 68.91 | 68.91 | 68.91 | 0.00 | 19.70 |
| LGA | 5 | 73.76 | 72.92 | 73.17 | 72.55 | 72.79 | 72.14 | 72.62 | 72.40 | 72.07 | 72.71 | 1.36 | 15.90 |
| LANDER | 5 | 66.90 | 65.63 | 65.13 | 63.62 | 63.33 | 62.53 | 63.78 | 64.78 | 64.15 | 64.43 | 2.12 | 24.18 |
| Fed-L2P | 5 | 73.47 | 77.66 | 79.49 | 81.02 | 82.53 | 83.81 | 83.53 | 84.19 | 84.78 | 81.17 | -10.72 | 7.44 |
| Fed-DualP | 5 | 76.39 | 85.00 | 87.19 | 87.50 | 87.89 | 87.60 | 87.98 | 87.86 | 88.09 | 86.17 | -11.47 | 2.44 |
| Fed-CODAP | 5 | 81.73 | 69.20 | 71.38 | 70.26 | 68.93 | 69.48 | 68.23 | 70.16 | 70.88 | 71.14 | 11.58 | 17.47 |
| Fed-Cprompt | 5 | 88.00 | 62.67 | 66.90 | 67.22 | 63.25 | 62.38 | 62.48 | 60.68 | 59.28 | 65.87 | 27.32 | 22.74 |
| UOPP | 5 | 90.57 | 90.57 | 90.56 | 90.57 | 90.57 | 90.56 | 90.57 | 90.56 | 72.97 | 88.61 | 0.01 | 0.00 |
| Fed-S3C | 1 | 44.51 | 49.42 | 49.96 | 49.52 | 49.56 | 49.21 | 48.97 | 48.93 | 48.06 | 48.68 | -4.42 | 40.68 |
| TARGET | 1 | 68.91 | 68.91 | 68.91 | 68.91 | 68.91 | 68.91 | 68.91 | 68.91 | 68.91 | 68.91 | 0.00 | 20.46 |
| LGA | 1 | 73.58 | 72.41 | 73.57 | 72.26 | 72.63 | 71.87 | 72.40 | 70.34 | 70.05 | 72.12 | 3.24 | 17.24 |
| LANDER | 1 | 66.90 | 65.43 | 64.12 | 63.10 | 62.65 | 59.60 | 63.57 | 62.83 | 61.95 | 63.35 | 4.07 | 26.01 |
| Fed-L2P | 1 | 77.00 | 78.36 | 79.67 | 80.19 | 80.59 | 81.52 | 81.00 | 80.93 | 80.33 | 79.95 | -3.93 | 9.41 |
| Fed-DualP | 1 | 78.66 | 86.60 | 88.35 | 87.74 | 87.82 | 87.88 | 87.57 | 87.70 | 87.11 | 86.60 | -9.04 | 2.76 |
| Fed-CODAP | 1 | 83.29 | 74.56 | 74.18 | 72.00 | 71.12 | 69.91 | 67.09 | 70.44 | 72.33 | 72.33 | 12.85 | 17.03 |
| Fed-Cprompt | 1 | 87.53 | 85.38 | 82.57 | 81.40 | 80.73 | 79.60 | 79.48 | 77.92 | 75.77 | 81.15 | 9.62 | 8.21 |
| UOPP | 1 | 90.65 | 90.65 | 90.65 | 90.66 | 90.66 | 89.48 | 88.82 | 87.97 | 84.72 | 89.36 | 2.68 | 0.00 |

Table A13: Novel classes accuracy of the consolidated algorithms in CIFAR100 dataset with 5-shot and 1-shot setting across 3 different seeded runs. $S$ indicates the number of shots for the few shot tasks, PD indicates the performance drop, and Gap indicates the gap between the respected method to our proposed method (UOPP).

| Method | S | Novel Classes Accuracy in each session (%) | | | | | | | | Avg | PD | Gap |
|---|---|---|---|---|---|---|---|---|---|---|---|---|
| | | 1 | 2 | 3 | 4 | 5 | 6 | 7 | 8 | | | |
| Fed-S3C | 5 | 49.80 | 37.67 | 31.00 | 27.85 | 26.31 | 25.70 | 25.16 | 24.48 | 31.00 | 25.32 | 61.92 |
| TARGET | 5 | 0.00 | 0.00 | 0.00 | 0.00 | 0.00 | 0.00 | 0.00 | 0.00 | 0.00 | 0.00 | 92.92 |
| LGA | 5 | 32.27 | 20.10 | 11.13 | 9.12 | 6.87 | 6.75 | 5.33 | 4.13 | 11.96 | 28.14 | 80.96 |
| LANDER | 5 | 0.00 | 0.00 | 0.00 | 0.00 | 0.00 | 0.00 | 0.00 | 0.00 | 0.00 | 0.00 | 92.92 |
| Fed-L2P | 5 | 32.67 | 36.67 | 35.29 | 35.82 | 39.29 | 40.77 | 42.04 | 43.76 | 38.29 | -11.09 | 54.63 |
| Fed-DualP | 5 | 55.80 | 60.43 | 54.02 | 56.03 | 55.84 | 57.22 | 58.33 | 60.64 | 57.29 | -4.84 | 35.63 |
| Fed-CODAP | 5 | 70.40 | 67.40 | 62.33 | 61.88 | 58.12 | 56.52 | 55.60 | 53.98 | 60.78 | 16.43 | 32.14 |
| Fed-Cprompt | 5 | 88.20 | 83.70 | 68.07 | 63.80 | 62.20 | 58.37 | 58.23 | 58.58 | 67.64 | 29.63 | 25.27 |
| UOPP | 5 | 90.73 | 92.60 | 92.56 | 93.20 | 93.77 | 93.53 | 93.75 | 93.18 | 92.92 | -2.45 | 0.00 |
| Fed-S3C | 1 | 40.07 | 27.53 | 23.22 | 19.67 | 18.27 | 17.60 | 17.49 | 16.52 | 22.55 | 23.55 | 62.74 |
| TARGET | 1 | 0.00 | 0.00 | 0.00 | 0.00 | 0.00 | 0.00 | 0.00 | 0.00 | 0.00 | 0.00 | 85.29 |
| LGA | 1 | 2.07 | 2.67 | 5.36 | 9.68 | 11.11 | 13.67 | 13.11 | 16.88 | 9.32 | -14.81 | 75.97 |
| LANDER | 1 | 0.00 | 0.00 | 0.00 | 0.00 | 0.00 | 0.00 | 0.00 | 0.00 | 0.00 | 0.00 | 85.29 |
| Fed-L2P | 1 | 33.47 | 45.10 | 51.24 | 52.02 | 56.65 | 57.67 | 60.66 | 62.63 | 52.43 | -29.16 | 32.86 |
| Fed-DualP | 1 | 65.87 | 67.07 | 67.02 | 66.85 | 66.48 | 67.98 | 67.46 | 68.15 | 67.11 | -2.28 | 18.18 |
| Fed-CODAP | 1 | 57.80 | 62.65 | 53.87 | 56.73 | 55.04 | 52.13 | 53.49 | 51.55 | 55.41 | 6.25 | 29.88 |
| Fed-Cprompt | 1 | 52.60 | 59.30 | 48.73 | 49.95 | 46.68 | 45.87 | 45.06 | 43.45 | 48.95 | 9.15 | 36.34 |
| UOPP | 1 | 84.27 | 85.90 | 84.82 | 86.15 | 85.43 | 85.34 | 85.46 | 84.96 | 85.29 | -0.69 | 0.00 |

# I  DETAILED NUMERICAL RESULTS ON STABILITY-PLASTICITY ANALYSIS

In this section, we present the detailed numerical results on the stability-plasticity analysis of UOPP as shown in Tables A12, A13, and A14.

# J  DETAILED NUMERICAL RESULTS DIFFERENT LOCAL CLIENTS AND GLOBAL ROUNDS

In this section we present the detailed numerical results on different local clients and rounds as presented in tables A15 and A16.

# K  DETAILED NUMERICAL RESULTS OF ABLATION STUDY

In this section we present detailed numerical results on the ablation study as shown in table A17.

Table A14: harmonic Mean accuracy of the consolidated algorithms in CIFAR100 dataset with 5-shot and 1-shot setting across 3 different seeded runs. $S$ indicates the number of shots for the few shot tasks, PD indicates the performance drop, and Gap indicates the gap between the respected method to our proposed method (UOPP).

| Method | S | Harmonic Mean Accuracy in each session (%) | | | | | | | | Avg | PD | Gap |
|---|---|---|---|---|---|---|---|---|---|---|---|---|
| | | 1 | 2 | 3 | 4 | 5 | 6 | 7 | 8 | | | |
| Fed-S3C | 5 | 49.35 | 42.76 | 37.95 | 35.43 | 33.93 | 33.39 | 32.91 | 32.08 | 37.23 | 17.26 | 53.27 |
| TARGET | 5 | 0.00 | 0.00 | 0.00 | 0.00 | 0.00 | 0.00 | 0.00 | 0.00 | 0.00 | 0.00 | 90.50 |
| LGA | 5 | 44.74 | 31.54 | 19.30 | 16.20 | 12.54 | 12.35 | 9.93 | 7.81 | 19.30 | 36.93 | 71.20 |
| LANDER | 5 | 0.00 | 0.00 | 0.00 | 0.00 | 0.00 | 0.00 | 0.00 | 0.00 | 0.00 | 0.00 | 90.50 |
| Fed-L2P | 5 | 45.99 | 50.18 | 49.16 | 49.95 | 53.50 | 54.79 | 56.08 | 57.72 | 52.17 | -11.74 | 38.33 |
| Fed-DualP | 5 | 67.37 | 71.39 | 66.80 | 68.44 | 68.20 | 69.34 | 70.12 | 71.83 | 69.19 | -4.46 | 21.31 |
| Fed-CODAP | 5 | 69.79 | 69.33 | 66.06 | 65.21 | 63.30 | 61.82 | 62.04 | 61.28 | 64.85 | 8.51 | 25.65 |
| Fed-Cprompt | 5 | 73.27 | 74.36 | 67.64 | 63.52 | 62.29 | 60.35 | 59.43 | 58.93 | 64.98 | 14.35 | 25.52 |
| PILoRA | 5 | 0.00 | 0.00 | 0.00 | 0.00 | 0.00 | 0.00 | 0.00 | 0.00 | 0.00 | 0.00 | 90.50 |
| UOPP | 5 | 90.65 | 91.57 | 91.55 | 91.86 | 92.14 | 92.03 | 92.13 | 81.85 | 90.47 | 8.80 | 0.00 |
| Fed-S3C | 1 | 44.25 | 35.50 | 31.62 | 28.16 | 26.64 | 25.90 | 25.77 | 24.58 | 30.30 | 19.67 | 56.90 |
| TARGET | 1 | 0.00 | 0.00 | 0.00 | 0.00 | 0.00 | 0.00 | 0.00 | 0.00 | 0.00 | 0.00 | 87.20 |
| LGA | 1 | 4.02 | 5.15 | 9.97 | 17.09 | 19.24 | 22.99 | 22.10 | 27.20 | 15.97 | -23.19 | 71.23 |
| LANDER | 1 | 0.00 | 0.00 | 0.00 | 0.00 | 0.00 | 0.00 | 0.00 | 0.00 | 0.00 | 0.00 | 87.20 |
| Fed-L2P | 1 | 46.90 | 57.60 | 62.53 | 63.22 | 66.85 | 67.37 | 69.34 | 70.38 | 63.02 | -23.48 | 24.18 |
| Fed-DualP | 1 | 74.82 | 76.25 | 76.00 | 75.91 | 75.70 | 76.54 | 76.26 | 76.47 | 75.99 | -1.65 | 11.21 |
| Fed-CODAP | 1 | 65.12 | 67.93 | 61.63 | 63.11 | 61.59 | 58.67 | 60.80 | 58.80 | 62.21 | 6.32 | 24.99 |
| Fed-Cprompt | 1 | 65.10 | 69.03 | 60.97 | 61.72 | 58.85 | 58.17 | 57.10 | 55.23 | 60.77 | 9.87 | 26.43 |
| PILoRA | 1 | 0.00 | 0.00 | 0.00 | 0.00 | 0.00 | 0.00 | 0.00 | 0.00 | 0.00 | 0.00 | 87.20 |
| UOPP | 1 | 87.34 | 88.21 | 87.64 | 88.35 | 87.41 | 87.05 | 86.69 | 84.84 | 87.19 | 2.50 | 0.00 |

Table A15: Accuracy of the consolidated algorithms in CIFAR100 dataset with 5-shot setting on different number of selected local clients across 3 different seeded runs. $S$ indicates the number of shots for the few shot tasks, PD indicates the performance drop, and L indicates the number of selected local clients.

| Method | L | Accuracy in each session (%) | | | | | | | | | Avg | PD |
|---|---|---|---|---|---|---|---|---|---|---|---|---|
| | | 0 | 1 | 2 | 3 | 4 | 5 | 6 | 7 | 8 | | |
| S3C | 4 | 42.63 | 50.02 | 48.77 | 46.47 | 44.56 | 42.72 | 41.53 | 40.39 | 38.39 | 43.94 | 4.24 |
| S3C | 6 | 44.51 | 48.97 | 47.77 | 45.35 | 43.48 | 41.47 | 40.33 | 39.32 | 37.71 | 43.21 | 6.80 |
| S3C | 8 | 43.43 | 48.60 | 46.90 | 44.81 | 43.16 | 41.34 | 40.33 | 38.84 | 37.21 | 42.74 | 6.22 |
| TARGET | 4 | 66.75 | 61.62 | 57.21 | 53.40 | 50.06 | 47.12 | 44.50 | 42.16 | 40.05 | 51.43 | 26.70 |
| TARGET | 6 | 68.90 | 63.61 | 59.06 | 55.12 | 51.68 | 48.64 | 45.94 | 43.52 | 41.34 | 53.09 | 27.56 |
| TARGET | 8 | 73.53 | 67.88 | 63.03 | 58.83 | 55.15 | 51.91 | 49.02 | 46.44 | 44.12 | 56.66 | 29.41 |
| LGA | 4 | 72.98 | 68.8 | 62.81 | 57.57 | 54.29 | 51.51 | 49.81 | 46.67 | 42.58 | 56.34 | 30.40 |
| LGA | 6 | 73.76 | 69.80 | 65.59 | 60.26 | 56.87 | 52.94 | 50.66 | 47.69 | 44.89 | 58.05 | 28.87 |
| LGA | 8 | 73.73 | 70.03 | 65.9 | 60.4 | 56.31 | 52.68 | 50.37 | 47.32 | 44.61 | 57.93 | 29.12 |
| LANDER | 4 | 59.60 | 58.03 | 53.23 | 49.81 | 45.80 | 43.11 | 40.64 | 39.06 | 37.27 | 47.40 | 22.33 |
| LANDER | 6 | 58.60 | 61.75 | 56.26 | 52.11 | 47.71 | 44.71 | 41.69 | 40.28 | 38.87 | 49.11 | 19.73 |
| LANDER | 8 | 61.60 | 63.80 | 58.84 | 54.37 | 50.46 | 47.76 | 44.17 | 42.18 | 40.82 | 51.56 | 20.78 |
| Fed-DualP | 4 | 64.90 | 75.14 | 80.11 | 78.63 | 79.35 | 77.79 | 76.84 | 76.67 | 76.05 | 76.17 | -11.15 |
| Fed-DualP | 6 | 76.39 | 82.75 | 83.37 | 80.80 | 79.93 | 78.26 | 77.73 | 76.98 | 77.11 | 79.26 | -0.72 |
| Fed-DualP | 8 | 84.65 | 84.77 | 83.90 | 80.41 | 78.56 | 76.64 | 75.93 | 74.60 | 73.58 | 79.23 | 11.07 |
| Fed-Cprompt | 4 | 87.78 | 44.85 | 35.84 | 35.13 | 38.63 | 40.78 | 38.12 | 41.14 | 41.30 | 44.84 | 46.48 |
| Fed-Cprompt t | 6 | 88.00 | 64.63 | 69.30 | 67.39 | 63.39 | 62.33 | 61.11 | 59.78 | 59.00 | 66.10 | 29.00 |
| Fed-Cprompt | 8 | 87.65 | 82.52 | 80.99 | 77.53 | 76.64 | 73.74 | 71.70 | 70.67 | 68.39 | 76.65 | 19.26 |
| UOPP | 4 | 89.18 | 89.49 | 89.57 | 89.91 | 90.35 | 89.99 | 89.23 | 88.88 | 84.96 | 89.06 | 4.22 |
| UOPP | 6 | 90.57 | 90.58 | 90.85 | 90.96 | 91.23 | 91.51 | 91.56 | 91.74 | 81.05 | 90.01 | 9.52 |
| UOPP | 8 | 90.93 | 90.91 | 91.40 | 91.61 | 91.74 | 91.78 | 91.26 | 91.36 | 90.90 | 91.32 | 0.03 |

Table A16: Accuracy of the consolidated algorithms in CIFAR100 dataset with 5-shot setting on different number of rounds across 3 different seeded runs. $S$ indicates the number of shots for the few shot tasks, PD indicates the performance drop, and R indicates the number of rounds.

| Method | R | Accuracy in each session (%) | | | | | | | | | Avg | PD |
|---|---|---|---|---|---|---|---|---|---|---|---|---|
| | | 0 | 1 | 2 | 3 | 4 | 5 | 6 | 7 | 8 | | |
| S3C | 54 | 43.42 | 49.51 | 48.01 | 45.41 | 43.69 | 41.96 | 40.89 | 39.64 | 38.10 | 43.40 | 5.32 |
| S3C | 72 | 51.20 | 53.95 | 51.97 | 49.09 | 47.10 | 45.11 | 43.74 | 42.88 | 41.13 | 47.35 | 10.07 |
| S3C | 90 | 44.51 | 48.97 | 47.77 | 45.35 | 43.48 | 41.47 | 40.33 | 39.32 | 37.71 | 43.21 | 6.80 |
| TARGET | 54 | 57.60 | 53.17 | 49.37 | 46.08 | 43.20 | 40.66 | 38.40 | 36.38 | 34.56 | 44.38 | 23.04 |
| TARGET | 72 | 67.28 | 62.11 | 57.67 | 53.83 | 50.46 | 47.49 | 44.86 | 42.49 | 40.37 | 51.84 | 26.91 |
| TARGET | 90 | 68.90 | 63.61 | 59.06 | 55.12 | 51.68 | 48.64 | 45.94 | 43.52 | 41.34 | 53.09 | 27.56 |
| LGA | 54 | 69.57 | 62.32 | 60.94 | 61.41 | 57.44 | 53.71 | 49.76 | 51.24 | 47.51 | 57.10 | 22.06 |
| LGA | 72 | 68.35 | 66.26 | 61.61 | 58.15 | 54.75 | 51.55 | 49.21 | 45.19 | 42.08 | 55.24 | 26.27 |
| LGA | 90 | 73.76 | 69.80 | 65.59 | 60.26 | 56.87 | 52.94 | 50.66 | 47.69 | 44.89 | 58.05 | 28.87 |
| LANDER | 54 | 60.60 | 43.89 | 40.59 | 38.16 | 34.89 | 32.74 | 31.04 | 30.14 | 29.23 | 37.92 | 31.37 |
| LANDER | 72 | 62.60 | 57.00 | 52.09 | 49.21 | 46.21 | 43.13 | 40.42 | 37.05 | 36.64 | 47.15 | 25.96 |
| LANDER | 90 | 58.60 | 61.75 | 56.26 | 52.11 | 47.71 | 44.71 | 41.69 | 40.28 | 38.87 | 49.11 | 19.73 |
| Fed-DualP | 54 | 79.40 | 81.40 | 83.44 | 82.67 | 82.39 | 81.85 | 81.31 | 80.78 | 80.32 | 81.51 | -0.92 |
| Fed-DualP | 72 | 78.85 | 83.25 | 83.96 | 81.61 | 80.28 | 79.73 | 78.63 | 78.15 | 77.06 | 80.17 | 1.79 |
| Fed-DualP | 90 | 76.39 | 82.75 | 83.37 | 80.80 | 79.93 | 78.26 | 77.73 | 76.98 | 77.11 | 79.26 | -0.72 |
| Fed-Cprompt | 54 | 87.92 | 72.05 | 64.70 | 54.99 | 66.09 | 52.84 | 56.98 | 55.86 | 51.01 | 62.49 | 36.91 |
| Fed-Cprompt | 72 | 87.92 | 72.42 | 49.91 | 57.41 | 60.11 | 55.87 | 53.98 | 49.23 | 49.84 | 59.63 | 38.08 |
| Fed-Cprompt | 90 | 88.00 | 64.63 | 69.30 | 67.39 | 63.39 | 62.33 | 61.11 | 59.78 | 59.00 | 66.10 | 29.00 |
| UOPP | 54 | 89.87 | 89.75 | 90.33 | 90.73 | 90.91 | 91.12 | 91.33 | 91.18 | 91.32 | 90.73 | -1.45 |
| UOPP | 72 | 90.37 | 90.29 | 90.70 | 90.99 | 90.94 | 91.34 | 91.27 | 91.07 | 90.13 | 90.79 | 0.24 |
| UOPP | 90 | 90.57 | 90.58 | 90.85 | 90.96 | 91.23 | 91.51 | 91.56 | 91.74 | 81.05 | 90.01 | 9.52 |

Table A17: Accuracy of different configurations in CIFAR100 dataset with 5-shot setting on across 3 different seeded runs. $S$ indicates the number of shots for the few shot tasks, PD indicates the performance drop, and Gap indicates the difference accuracy to PIP.

| Conf. | Accuracy in each session (%) | | | | | | | | | Avg | PD | Gap |
|---|---|---|---|---|---|---|---|---|---|---|---|---|
| | 0 | 1 | 2 | 3 | 4 | 5 | 6 | 7 | 8 | | | |
| A (w/o Static Proto) | 84.37 | 82.54 | 80.91 | 80.56 | 80.29 | 80.54 | 80.70 | 79.03 | 76.61 | 80.62 | 7.76 | 9.39 |
| B (w/o Dynamic Proto) | 90.27 | 87.38 | 85.67 | 84.40 | 84.69 | 84.84 | 85.24 | 85.16 | 80.21 | 85.32 | 10.06 | 4.69 |
| C (w/o MLP Head) | 88.25 | 88.66 | 89.07 | 89.16 | 89.63 | 90.11 | 90.27 | 90.62 | 82.76 | 88.72 | 5.49 | 1.29 |
| D (w/o PB. Head) | 90.10 | 83.17 | 77.23 | 72.08 | 67.58 | 63.60 | 60.07 | 56.91 | 52.34 | 69.23 | 37.76 | 20.78 |
| UOPP | 90.57 | 90.58 | 90.85 | 90.96 | 91.23 | 91.51 | 91.56 | 91.74 | 81.05 | 90.01 | 9.52 | 0.00 |

