# OpenReview forum: "Federated Few-Shot Class-Incremental Learning"
_ICLR.cc/2025/Conference — ICLR 2025 Poster_

### Official Review · Reviewer_13iY · 2024-11-02

**Soundness:** 4
**Presentation:** 4
**Contribution:** 3
**Rating:** 8
**Confidence:** 4

**Summary:**

This paper define a new Federated Few-shot Class-Incremental Learning (FFSCIL) problem which is interesting and has significante value in real model deployment. A novel Univfied Optimizaed Prototype Prompt (UOPP) method is proposed for FFSCIL problem. UOPP utilizes task-wise prompt-learning to mitigate task interference empowered by shared static-dynamic prototypes, adaptive dual heads, and weighted aggregation. The dynamic prototype tackles prototype bias by iterative rectiffcations. Comprehensive experimental results show that UOPP signiffcantly outperforms existing SOTA methods of FCIL, FSCIL, and CIL, on three benchmarks with a signiffcant gap. The proposed method achieves better stability-plasticity trade-off, and robustness in different local clients and small global rounds.

**Strengths:**

1. A new Federated Few-shot Class-Incremental Learning (FFSCIL) problem which is interesting and has significant value in real model deployment.
2. An effective method UOPP is proposed for FFSCIL.
3. Comprehensive experiments are designed and demonstrate the effectiveness in final performance, stability-plasticity balance, and robustness, which are important in class-incremental learning.
4. The whole paper is well organized and easy to follow. The problem and idea are clearly presented.
5. Codes are available and theoretical analysis is provided.

**Weaknesses:**

1. The few-shot constraint when T>0 is defined in the task level or just client level is vague. Could the authors explain whether the images assigned to each client is randomly selected from the whole training set or from a small fixed subset of the training images? If former manner, the few-shot constraint is defined in the client level and prototype rectification through different clients is able to improve the performance. If the latter manner, the few-shot constraint is defined in the task-level. Prototype rectification through globally stored representation is limited to improve the learned prototype as no more information provided.
2. Unclear details: How Z_l is augmented to construct query sample set Q? What's the loss to train GradNet? How is ω_l determined in server-side aggregation? How is setting of D implemented in table 4?
3. Typos: line270, "parameter Ψ" should be "parameter Φ". line362: "10 round per class" should be "10 round per task".

**Questions:**

1. Is there any error in equation (4)? As ω is a regularizer not a weight, should ωzc(s) be ω(zc(s))?
2. Why larger round leads to worse performance for UOPP according to table 3?

---

> ### Author Response · Authors · 2024-11-22
> **Author Response to Reviewer 13iY**
>
> W1. The few-shot constraint when $T>0$ is defined in the task level or just client level is vague. Could the authors explain whether the images assigned to each client is randomly selected from the whole training set or from a small fixed subset of the training images? If former manner, the few-shot constraint is defined in the client level and prototype rectification through different clients is able to improve the performance. If the latter manner, the few-shot constraint is defined in the task-level. Prototype rectification through globally stored representation is limited to improve the learned prototype as no more information provided.
>
> Our Response: Thank you for your concern. We have revised the problem formulation in our paper. We have applied the setting as in the earlier manner. Following a real practice scenario, each client most likely holds different samples from the other client for the same class e.g. a hospital has different patients (samples) from other hospitals for the same disease.
>
> W2. Unclear details: How $Z_l$ is augmented to construct query sample set Q? What's the loss to train GradNet? How is $w_l$ determined in server-side aggregation? How is setting of D implemented in table 4?
>
> Our Response: Thank you for your concern.
>
> - The query sample Q is constructed by drawing prototypes from a normal distribution \mathcal{N}(\mu_l,\sigma_l^2) where (\mu_l,\sigma_l^2)  is the property of z_l \in $Z_l$ . We have revised the description of this drawing process in our revised manuscript i.e. in the latest part of section 4.3.
>
> - We consider client participation on the current task as client weight $w_l$. We consider that more participation implies better weight and prototypes from a client, thus contributing more to the global model and prototypes. We mention it in section 4.5.
>
> W3. Typos: line270, "parameter $\psi$" should be "parameter $\Phi$". line362: "10 round per class" should be "10 round per task".
>
> Our Response: Thank you for your correction. We have revised the typo in our latest manuscript.
>
>
> Questions:
> Q1. Is there any error in equation (4)? As $\omega$ is a regularizer not a weight, should $\omega z_c(s)$ be $\omega(z_c(s))$?
>
> Our Response: Thank you for your correction. Yes, It is a typo. We have revised the typo in our latest manuscript.
>
>
> Why larger round leads to worse performance for UOPP according to table 3?
>
> Our Response: Thank you for your concern. A bigger round means more update processes on a model that may lead to overfitting. In our case, It causes too many updates on the task key that cause false task prediction on several samples. The use of early stopping or decreased learning rate may solve this issue as we discussed in Appendix E.

---

> > ### Comment · Reviewer_13iY · 2024-11-22
> >
> > Thanks for the response. I have no more question and will maintain my rating.

---

> > > ### Author Response · Authors · 2024-11-22
> > > **Author Response to Reviewer 13iY**
> > >
> > > We appreciate Reviewer 13iY's fast response and effort in reviewing our revised manuscript. We thank Reviewer 13iY for maintaining the score for our paper.

---

### Official Review · Reviewer_b8Za · 2024-11-02

**Soundness:** 3
**Presentation:** 3
**Contribution:** 3
**Rating:** 6
**Confidence:** 4

**Summary:**

The paper propose a new setting - Federated Few-Shot Class Incremental Learning (FFSCIL) where each client in the system has very few samples per class. The work focus on the prototype bias rectification process and propose a prompt-learning method named “Unified Optimized Prototype Prompt (UOPP)” which is based on static and dynamic class prototypes optimized using Neural Ordinary Differential Equations. The proposed method outperforms existing methods across three benchmark datasets and achieves better stability-plasticity trade-off.

**Strengths:**

1. The paper has good analysis of the prototype rectification problem with good illustrations.
2. The paper proposed a new and interesting few-shot setting for FCIL.
3. The method section is well-written with lot of theoretical analysis.
4. The extensive experiments and ablation study is appreciated.

**Weaknesses:**

1. The writing needs to be improved and more precise with appropriate and more FSCIL references. For instance, more context and details are necessary here (line 54 - “this mechanism may violate data privacy principles since it may reveal partial information about private data of a client to another party”; line 57 - “that may breach data openness policy, where data are only open for a client at a specific moment”). The introduction section could better introduce and establish the problem statement instead of directly jumping to the solution. The motivation for the proposed FFSCIL setting, why not save exemplars and why not use synthetic samples is not clear and needs to be discussed in details.

2. Existing works [1,2,3] which extensively studied prototype calibration for FSCIL which is the same concept termed as prototype rectification in this work, are completely ignored. TEEN [1] and statistics calibration [3] are very relevant single-step prototype rectification baselines which should be considered, adapted for FFSCIL and compared to the proposed method since this naive averaging method is already established for FSCIL. It is important to discuss the difference of class-similarity based rectification as in [1,3] with the proposed Neural ODE-based rectification and see its impact.

3. Recent work from Federated Learning [4] used prototype-based NCM classifiers and achieved very good performance without any training. Using NCM classifier is pretty common and standard practice in FSCIL where the model is trained in the base task and the frozen model is used in all subsequent tasks to compute class means for NCM classifier. FedNCM [4] is an important baseline method missing from the discussion and comparison tables. FedNCM is the baseline which shows how much the models can perform without any further fine-tuning or learning prompts after base task.

4. It is not clear what is the balance mean accuracy? Recent FSCIL papers [1,2,3] considered the harmonic mean accuracy which is quite helpful in FSCIL settings to evaluate performance of old and new classes since most of the accuracy comes from the old classes.


[1] Wang, Qi-Wei, et al. "Few-shot class-incremental learning via training-free prototype calibration." Advances in Neural Information Processing Systems 36 (2023).

[2] Peng, Can, et al. "Few-shot class-incremental learning from an open-set perspective." European Conference on Computer Vision, 2022.

[3] Goswami, Dipam, et al. "Calibrating Higher-Order Statistics for Few-Shot Class-Incremental Learning with Pre-trained Vision Transformers." Proceedings of the IEEE/CVF Conference on Computer Vision and Pattern Recognition. 2024.

[4] Legate, Gwen, et al. "Guiding the last layer in federated learning with pre-trained models." Advances in Neural Information Processing Systems 36 (2023).

**Questions:**

1. It is not clear if the ViT model used for UOPP in the experiments used pre-trained weights or not. This is not mentioned in the paper. It is important to have a fair comparison and start from the same weights for all compared methods.

---

> ### Author Response · Authors · 2024-11-22
> **Author Response to Reviewer b8Za for W1-W4**
>
> W1. The writing needs to be improved and more precise with appropriate and more FSCIL references. For instance, more context and details are necessary here (line 54 - “this mechanism may violate data privacy principles since it may reveal partial information about private data of a client to another party”; line 57 - “that may breach data openness policy, where data are only open for a client at a specific moment”). The introduction section could better introduce and establish the problem statement instead of directly jumping to the solution. The motivation for the proposed FFSCIL setting, why not save exemplars and why not use synthetic samples is not clear and needs to be discussed in details.
>
> Our Response: Thank you for your concern. We have revised our introduction section to improve the understanding of our motivation and the limitations of existing methods.
>
> W2. Existing works [1,2,3] which extensively studied prototype calibration for FSCIL which is the same concept termed as prototype rectification in this work, are completely ignored. TEEN [1] and statistics calibration [3] are very relevant single-step prototype rectification baselines which should be considered, adapted for FFSCIL and compared to the proposed method since this naive averaging method is already established for FSCIL. It is important to discuss the difference of class-similarity based rectification as in [1,3] with the proposed Neural ODE-based rectification and see its impact.
>
> Our Response: Thank you for your concern.
>
> - Because of time constraints, we can't directly adapt the FSCIL methods into the FFSCIL setting. However,  we have compared our method to the mentioned SOTA as well as the recent SOTAs in the standalone (single client) FSCIL setting. Please see our analysis in Appendix D.1. Our analysis shows that our method achieves better performance than the SOTAs with a significant margin. It confirms the effectiveness of our proposed method in the case of a standalone setting where the FSCIL is conducted by a single centralized client.
>
> - The main difference between our method and the mentioned methods [1,3] are as follows: (1) From a prototype rectifications perspective, our method utilizes a trainable network i.e. Neural ODEs that work based on query and support samples (prototypes) while the existing methods utilize a ratio of novel classes prototype similarity and base classes prototype similarity. In our method, the query and support samples are drawn from different distributions i.e. local and global respectively.  (2) Our prototype utilizes an adaptive dual-head classifier while the existing methods don't.  (3) Our method has been proven in a federated setting where some classes may not be available in a local client, while the existing methods don't. We have emphasized the uniqueness of our method in section 4, in the overview part before section 4.1.
>
>
>
> W3. Recent work from Federated Learning [4] used prototype-based NCM classifiers and achieved very good performance without any training. Using NCM classifier is pretty common and standard practice in FSCIL where the model is trained in the base task and the frozen model is used in all subsequent tasks to compute class means for NCM classifier. FedNCM [4] is an important baseline method missing from the discussion and comparison tables. FedNCM is the baseline which shows how much the models can perform without any further fine-tuning or learning prompts after base task.
>
> Our Response: Thank you for your concern. At this very limited time, we can't compare FedNCM in the FFSCIL  setting since FedNCM was not proposed for class incremental learning or few-shot class incremental learning setting. However, We have discussed the potential solution of FedNCM in our intended literature review, please see in Appendix E.
>
> W4. It is not clear what is the balance mean accuracy? Recent FSCIL papers [1,2,3] considered the harmonic mean accuracy which is quite helpful in FSCIL settings to evaluate performance of old and new classes since most of the accuracy comes from the old classes.
>
> Our Response: Thank you for your advice. We have replaced our stability-plasticity metrics into harmonic mean accuracy. Previously, we defined balance mean accuracy as simple average of base classes accuracy and novel classes accuracy i.e. Acc(Balance)=(Acc(BaseClasses)+Acc(NovelClasses))/2

---

> ### Author Response · Authors · 2024-11-22
> **Author Response to Reviewer 2VKj for W1/Q1**
>
> Q1. It is not clear if the ViT model used for UOPP in the experiments used pre-trained weights or not. This is not mentioned in the paper. It is important to have a fair comparison and start from the same weights for all compared methods.
>
> Our Response: Thank you for your question. Following the standard practice in parameter efficient fine-tuning method, all the prompt-based and PILoRA methods utilized pre-trained ViT. For the other methods, we have followed the advised settings from their official papers and code implementations. By doing so, we believe that we maintain fairness in our experiments. We describe the detailed experimental settings in Appendix F.

---

> > ### Comment · Reviewer_b8Za · 2024-11-22
> >
> > I am satisfied with most of the author's responses and appreciate the efforts to address most of the questions and weaknesses. The paper in the current version looks good to me with a lot of additional clarifications, motivation, some experiments, analysis, comparisons and robust metrics.
> >
> > I have updated my score to 6.

---

> ### Author Response · Authors · 2024-11-22
> **Author Response to Reviewer b8Za**
>
> We appreciate Reviewer b8Za's time and effort in reviewing our revised manuscript at a timely speed. We thank Reviewer b8Za for increasing the score for our paper.

---

### Official Review · Reviewer_2VKj · 2024-11-03

**Soundness:** 3
**Presentation:** 3
**Contribution:** 3
**Rating:** 6
**Confidence:** 3

**Summary:**

The authors introduce Federated Few-Shot Class-Incremental Learning (FFSCIL), a new federated learning framework, along with the Unified Optimized Prototype Prompt (UOPP) model. UOPP tackles catastrophic forgetting, overfitting, and prototype bias by integrating task-specific prompt learning, unified static-dynamic prototypes, and an adaptive dual-head structure. Experiments demonstrate that UOPP surpasses state-of-the-art methods across multiple datasets, showing strong performance and robustness with low communication costs and efficient runtime.

**Strengths:**

•	Innovative Approach: UOPP combines prompt learning, prototype rectification, and an adaptive dual-head structure to address key challenges like catastrophic forgetting and overfitting.
•	Comprehensive Experiments: Extensive experiments demonstrate UOPP's robustness and superior performance across multiple datasets.
•	Open Source Code: The provided source code enhances reproducibility and supports further research.

**Weaknesses:**

•	In what situations is Federated Few-Shot Class-Incremental Learning (FFSCIL) actually useful, and why is it important to have a specific approach for these cases?
•	The paper’s prototype rectification method might not fully solve prototype bias issues, especially when data is really limited. With few local samples, it may still struggle to perform well in non-i.i.d. settings.
•	The experimental setup could be a bit limited since many of the comparison methods don’t fully match the FFSCIL scenario, which might make the performance comparisons less relevant and not show the method’s true strengths.

**Questions:**

see weaknesses

---

> ### Author Response · Authors · 2024-11-21
> **Author Response to Reviewer 2VKj for W1/Q1**
>
> Q1/W1 In what situations is Federated Few-Shot Class-Incremental Learning (FFSCIL) actually useful, and why is it important to have a specific approach for these cases? • The paper’s prototype rectification method might not fully solve prototype bias issues, especially when data is really limited. With few local samples, it may still struggle to perform well in non-i.i.d. settings. • The experimental setup could be a bit limited since many of the comparison methods don’t fully match the FFSCIL scenario, which might make the performance comparisons less relevant and not show the method’s true strengths.
>
> Our Response: Thank you for your concern.
>
> - FFSCIL is useful when a distributed system (collection of agents) develops a deep learning model for a dynamic environment with data privacy constraints and data scarcity issues. The agents continually observe the environment, so that the learned classes keep increasing, but the new classes only hold small (few) samples, and the agents aren't allowed to share their data. These conditions are practically found in real practice i.e. several hospitals work together to develop automated recognition of some diseases. After some time, new variants (sup-types) of the disease have been found, but only a few samples (patients) are available. The hospitals can't share the patients' data due to privacy issues, but they need to adjust the recognition model w.r.t newly found variants. In that case, the best model development strategy is by using the federated few-shot class incremental learning setting.
>
> - We have added a new analysis for the performance of the consolidated methods in different non-i.i.d levels as presented in section 6.3.a The analysis shows that our method outperforms existing SOTAs in various non-i.i.d levels with a significant margin i.e. up to 40\%. Our further analysis in section 6.3.d shows the validation loss of novel classes with and without rectification. Our analysis shows that rectification reduces the prototype bias significantly.
>
> - To the best of our knowledge, FFSCIL is a new problem, and almost no existing method specifically developed for this problem. Thus, we compare our method with the SOTAs of federated class incremental learning (FCIL) and few-shot class incremental learning (FSCIL). In addition, we add new baseline algorithms in the revised version of the paper where our method still outperforms them with significant gaps.

---

### Official Review · Reviewer_2sKT · 2024-11-04

**Soundness:** 3
**Presentation:** 3
**Contribution:** 2
**Rating:** 5
**Confidence:** 4

**Summary:**

The paper defines a new problem named Federated Few-Shot ClassIncremental Learning (FFSCIL) problem, where clients only hold very few samples for new classes. They develop a novel Unified Optimized Prototype Prompt (UOPP) model that utilizes task-wise prompt learning to mitigate task interference empowered by shared static-dynamic prototypes, adaptive dual heads, and weighted aggregation. The proposed UOPP significantly outperforms existing SOTA methods of FCIL, FSCIL, and CIL, on three datasets, along with deeper analysis confirms that the proposed method achieves better stability-plasticity trade-off, and robustness in different local clients and small global rounds.

**Strengths:**

1. The experiments are comprehensive and solid, achieving state-of-the-art performance.
2. The paper includes a deeper theoretical analysis, demonstrating a better stability-plasticity trade-off and robustness across different local clients and smaller global rounds.
3. Resource analysis is thorough, covering complexity, running time, parameters, and communication costs.
4. Clear implementation details and algorithm pseudocode are provided, along with an anonymous GitHub repository, making reproduction accessible.

**Weaknesses:**

1.The motivation is very weak
The introduction of this work is poorly written; I cannot grasp the new insights regarding the federated learning task provided by the authors. The two components of this work (1) prototype calibration and (2) introduction of prompts have already been discussed in a large number of prototype-based continual learning and prompt-based continual learning works. The authors should emphasize the uniqueness of their work.

2.The lacked comparison with the recent Federated Class-Incremental Learning work
[1] also uses Parameter-Efficient Fine-Tuning technique for Federated Class-Incremental Learning, i.e., LoRA. Thus, except for the task difference, what is the true difference between the prompt used in this paper and the LoRA used in [1]? In addition, the difference about the prototpye use between this work and [1] should also introduced.
[1]PILoRA: Prototype Guided Incremental LoRA for Federated Class-Incremental Learning
ECCV 2024

3.The comparison with prototype works in continual learning is insufficient
Prototype Calibration is one of the important techniques in continuallearning, and there is a lot of existing work focusing on it, e.g., [2,3,4]. However, the authors do not compare their work with these efforts.
[2] Few-Shot Class-Incremental Learning via Training-Free Prototype Calibration, ECCV 2024
[3] Exemplar-free Continual Representation Learning via Learnable Drift Compensation, ECCV 2024
[4] Few-Shot Class-Incremental Learning via Training-Free Prototype Calibration, ICCV 2023

4.The possible data leakage issue
Although the authors do not use rehearsal to correct prototypes, the support set S is needed. Essentially, samples in the supported set S are pseudo rehearsal sampled from Gaussian distributions located on prototypes, and this technique is also common in continual learning, e.g., HiDe-Prompt [5]. I am concerned that the pseudo rehearsal still violates data privacy principles.
[5] Hierarchical Decomposition of Prompt-Based Continual Learning: Rethinking Obscured Sub-optimality.

**Questions:**

1. Please address the problems proposed in weaknesses.
2. Table captions should be placed at the top of each table according to the ICLR template guidelines.
3. To comprehensively evaluate the limitations in few-shot scenarios, it would be beneficial to explore additional settings between one-shot and five-shot, as well as settings with more than five shots. (optinal)
4. Although the method claims lower resource consumption, the running time of 6.05 hours is longer than that of most other methods.

---

> ### Author Response · Authors · 2024-11-21
> **Author Response to Reviewer 2sKT for W1-Q2**
>
> W1.The motivation is very weak The introduction of this work is poorly written; I cannot grasp the new insights regarding the federated learning task provided by the authors. The two components of this work (1) prototype calibration and (2) introduction of prompts have already been discussed in a large number of prototype-based continual learning and prompt-based continual learning works. The authors should emphasize the uniqueness of their work.
>
> Our Response: Thank you for your concern. We have elaborated our introduction to present better motivation and insights regarding federated few-shot class incremental learning and our solution. We explain the uniqueness of our method in sec 4 (before sec. 4.1)
>
> W2.The lacked comparison with the recent Federated Class-Incremental Learning work [1] also uses Parameter-Efficient Fine-Tuning technique for Federated Class-Incremental Learning, i.e., LoRA. Thus, except for the task difference, what is the true difference between the prompt used in this paper and the LoRA used in [1]? In addition, the difference about the prototpye use between this work and [1] should also introduced. [1]PILoRA: Prototype Guided Incremental LoRA for Federated Class-Incremental Learning ECCV 2024.
>
> Our Response: Thank you for your concern. We have added PILoRA as a new competitor in our experiment and analysis. Please note that PiLORA doesn't' release code and setting for mini-magnet and CUB dataset. Thus, we run PILoRA in CIFAR100 dataset only. It is seen that our method beats PiLORA by significant margins. Note that PiLORA does not yet consider the case of federated few-shot continual learning as ours, so the issue of prototype bias due to very few samples remains unexplored in PiLORA.
>
> W3.The comparison with prototype works in continual learning is insufficient Prototype Calibration is one of the important techniques in continuallearning, and there is a lot of existing work focusing on it, e.g., [2,3,4]. However, the authors do not compare their work with these efforts. [2] Few-Shot Class-Incremental Learning via Training-Free Prototype Calibration, ECCV 2024 [3] Exemplar-free Continual Representation Learning via Learnable Drift Compensation, ECCV 2024 [4] Few-Shot Class-Incremental Learning via Training-Free Prototype Calibration, ICCV 2023
>
> Our Response: Thank you for your concern. We have compared our method to the recommended SOTAs as well as the recent SOTAs in the standalone (single client) FSCIL setting since the methods are not proposed for federated settings. Please see our analysis in Appendix D.1. Our analysis shows that our method achieves better performance than the SOTAs with a significant margin. It confirms the effectiveness of our proposed method in the case of a standalone setting where the FSCIL is conducted by a single centralized client. Note that these SOTA algorithms adopt different prototype rectification mechanisms compared to ours with the Neural ODE concept.
>
>
> W4.The possible data leakage issue Although the authors do not use rehearsal to correct prototypes, the support set S is needed. Essentially, samples in the supported set S are pseudo rehearsal sampled from Gaussian distributions located on prototypes, and this technique is also common in continual learning, e.g., HiDe-Prompt [5]. I am concerned that the pseudo-rehearsal still violates data privacy principles. [5] Hierarchical Decomposition of Prompt-Based Continual Learning: Rethinking Obscured Sub-optimality.
>
> Our Response: Thank you for your concern. The support set S in Neural ODE is drawn from the augmented local prototypes, the query set Q is drawn from the augmented (shared) unified prototype. Please note that the prototypes are the end product of ViT which is not a reversible function i.e. we can construct the input images from the prototypes. In addition, the dimensions and ranges of the images and the prototypes are totally different i.e. SxSx3 (e.g. 224x224x3) vs D(e.g. 768). Thus, the prototype sharing doesn't reveal any information about the client data and violates the data privacy issue. To avoid pseudo-rehearsal, we draw Q from currently learned class prototypes only. We can scale up the standard deviation i.e. $\sigma_{l}$' = r . $\sigma_{l}$ to obtain a wider range of query samples Q, where $r>1$ is the scale-up ratio, or we can assume that the prototype has stdev=1 following a normal distribution.
> However, looking at prototype-based FSCIL methods, all of them utilize base task prototypes in few-shot tasks, whether they augment them or not.  In that case, we need a clear boundary between rehearsal and rehearsal-free. Nevertheless, we believe that the prototypes don't reveal the original data (images) as they are produced by irreversible function.
>
> Q2. Table captions should be placed at the top of each table according to the ICLR template guidelines.
>
> Our response: Thank you for your concern. We have repositioned the table captions on top of the table in our revised manuscript.

---

> ### Author Response · Authors · 2024-11-21
> **Author Response to Reviewer 2sKT for Q3-Q4**
>
> Q3. To comprehensively evaluate the limitations in few-shot scenarios, it would be beneficial to explore additional settings between one-shot and five-shot, as well as settings with more than five shots. (optional)
>
> Our response: Thank you for your concern. We have added a new analysis of the performance of consolidated methods in advised k-shot settings i.e. 3-shot and 7-shot. Please see the analysis in Appendix D.2.
>
> Q4. Although the method claims lower resource consumption, the running time of 6.05 hours is longer than that of most other methods.
>
> Our response: Thank you for your concern.
>
> - Our running time is still lower than LGA, comparable to PILoRA, and higher than TARGET, LANDER, and Fed-Prompt-based methods. Compared to the Prompt-based, our method has an additional process i.e. prototype rectification that contributes significantly to the total simulation time.  Please note that the mentioned simulation time is computed from a simulation on a single GPU where all the clients and the server are executed sequentially. In practice, the difference should not be that big since all clients' local training is conducted in parallel across different devices. The other factor is the use of smaller local training epochs i.e. by LANDER, TARGET, Fed-CODAP, and Fed-CPrompt.
>
> - However, in terms of communication costs between client and server, our methods
> require one of the smallest costs. It is an important consideration for federated settings where the server interacts with multiple clients in each round.

---

> ### Author Response · Authors · 2024-11-26
> **Updated Response to Reviewer 2sKT for W4**
>
> After rechecking our implementation,  We confirm that our method has already satisfied with the rehearsal-free policy since query sample $\mathcal{Q}$ is drawn only from shared dynamic prototype $\hat{Z_l}$ that contains prototypes from the currently learned class, not the whole $Z_l$ that contains previously learned classes as well. Thus, we confirm our reported experiment results are obtained in a rehearsal-free manner. Please note that $Z_l=\tilde{Z}_l \cup \hat{Z}_l$ is assigned by aggregated prototype i.e. $Z_l = Z_G=\tilde{Z}_G \cup \hat{Z}_G$ on each federated round.
>
> We have updated our explanation in the main paper i.e. in the last paragraph of section 4.3.

---

> ### Author Response · Authors · 2024-11-30
> **Follow Up**
>
> Dear Reviewer 2sKT,
>
> Thank you for the constructive and insightful feedbacks. We would like to follow up on our response to your concerns.
> Could you please to review our updated paper.
>
> Here are the pointers of our revisions, addressing your concerns:
>
>
> W1. Stronger Motivation and uniqueness of the proposed method.
>
> - Section 1 : Stronger motivation on federated few-shot class incremental learning
>
> - Section 4 : Overview and uniqueness of our method
>
>
> W2. Comparison to recent FCIL SOTA i.e. PILoRA
>
> - Section 6.2-6.3, Tables 1,2,3, and 5, figure 4-5: PILoRA is included in our comparison, in all scenarios.
>
>
> W3. Comparison to recent FSCIL SOTAs
>
> - Appendix D.1: Comparison in standalone FSCIL, compared to recent FSCIL SOTAs
>
> W4. Data Leakage and pseudo-rehearsal issues:
>
> - Section 4.2 : How to generate prototypes
>
> - Our comment: Prototype space is totally different from input space both in terms of format and data ranges, ViT is non reversed function.
>
> - Section 4.3 last, paragraph: Our method complies non rehearsal and non pseudo-rehearsal policy, since $\mathcal{Q}$ is drawn from $\hat{Z}_l$ only, not $Z_l$ as a whole.
>
>
> Q2. Table caption:
> - Our updated manuscript: Table caption is located at the top of respective table
>
> Thank you
>
> Best Regards,
>
> Author of Submission 6715

---

### Official Review · Reviewer_KvkJ · 2024-11-04

**Soundness:** 3
**Presentation:** 4
**Contribution:** 4
**Rating:** 6
**Confidence:** 4

**Summary:**

The paper introduces a novel model named Unified Optimized Prototype Prompt (UOPP) to address the Federated Few-Shot Class-Incremental Learning (FFSCIL) problem. This problem is characterized by clients holding very few samples for new classes, which presents challenges such as catastrophic forgetting, overfitting, and prototype bias. The UOPP model incorporates task-wise prompt learning, unified static-dynamic prototypes optimized by Neural Ordinary Differential Equations (ODEs), and adaptive dual heads for enhanced inference. The dynamic prototypes are particularly designed to rectify prototype bias, which is a significant issue in few-shot learning scenarios. It defines the Federated Few-Shot Class-Incremental Learning (FFSCIL) problem and proposes the UOPP model to address it. UOPP uses prompt learning and dynamic prototypes to handle task interference, overfitting, and prototype bias. Experiments show UOPP outperforms existing methods on three datasets, with up to 76% higher accuracy and 67% better balance mean accuracy. It provides theoretical studies on the convergence and generalization of the UOPP method. It shows that UOPP is robust under various numbers of local clients and global rounds, with low communication costs and moderate running time.

**Strengths:**

1. The paper presents a novel solution for the challenging Federated Few-Shot Class-Incremental Learning (FFSCIL) problem, combining few-shot learning with federated learning. The UOPP model addresses this by integrating prompt learning, dynamic prototypes, and a dual-head classifier, showing high effectiveness.

2. The UOPP model excels on benchmark datasets, surpassing existing methods and proving its real-world applicability. It's supported by a robust theoretical framework that ensures convergence and generalization.

3. The UOPP model's utilization of dynamic prototype rectification through Neural Ordinary Differential Equations (ODEs) is a significant technical strength. This approach allows for the continuous optimization of prototypes, which are critical for representing new classes in few-shot learning scenarios.

4. Designed with practical considerations in mind, UOPP respects data privacy, optimizes communication efficiency, and accommodates computational constraints, making it an ideal solution for federated learning environments.

5. Enhancing transparency and fostering further research, the paper makes the UOPP source code publicly available.

**Weaknesses:**

1.The paper lacks a discussion on the future outlook and limitations of the method, which could provide a more comprehensive understanding of the research's implications, potential areas for advancement, and the constraints that may affect its practical application in various scenarios.

2 In federated learning, there's always a risk of introducing bias due to non-i.i.d. data distribution across clients, which the paper might not fully address.

3. There is a need for a more detailed analysis on how the model's performance is affected over extended periods and with a growing number of tasks, including the sustainability of its memory retention and its ability to adapt to new data without significant decreases in accuracy on previously learned tasks.

4. Although the model aims to be efficient, the use of advanced techniques like Neural ODEs could still demand significant computational resources, potentially limiting its applicability in resource-constrained environments. The paper does not sufficiently discuss this trade-off, which could be a critical consideration for practical deployment scenarios.

**Questions:**

1. How does the scalability of the UOPP model hold up when there is a significant increase in the number of clients or data volume? Can the model adapt to the continuous addition of new categories without a significant increase in training time or resource consumption?

2. How does the UOPP model perform in the face of class imbalance or distribution shift? Has the model undergone tests for generalization capabilities across different domains or tasks?

3. What specific operations are used in the iterative correction process of dynamic prototypes to guide them closer to the actual data distribution? How is the number of iterations for prototype correction determined? Is there an early stopping mechanism to prevent overfitting?

4. Regarding the dynamic updating of prototypes, can Neural ODEs be replaced by other optimization strategies, or how significant is the impact on the results without the ODE method, and was its computational complexity considered?

---

> ### Author Response · Authors · 2024-11-21
> **Author Response to Reviewer KvKJ for W1-W4**
>
> W1.The paper lacks a discussion on the future outlook and limitations of the method, which could provide a more comprehensive understanding of the research's implications, potential areas for advancement, and the constraints that may affect its practical application in various scenarios.
>
> Our Response: Thank you for your concern. We have added a discussion regarding the limitations of our method and potential solutions for them as presented in Appendix E.
>
> W2. In federated learning, there's always a risk of introducing bias due to non-i.i.d. data distribution across clients, which the paper might not fully address.
>
> Our Response: Thank you for your concern. We have added a new analysis for the performance of the consolidated methods across different non-i.i.d levels as presented in section 6.3. Our analysis shows that our method outperforms existing SOTAs in various non-i.i.d levels with a significant margin i.e. up to 40\%. Our further analysis is in section 6.3.a. shows the validation loss of novel classes with and without rectification. The analysis shows the rectification reduces the prototype bias significantly.
>
>
> W3. There is a need for a more detailed analysis on how the model's performance is affected over extended periods and with a growing number of tasks, including the sustainability of its memory retention and its ability to adapt to new data without significant decreases in accuracy on previously learned tasks.
>
> Our Response: Thank you for your concern.
>
> - We have discussed the performance of the method w.r.t. the growing number of tasks in stability-plasticity analysis (section 6.2.b). In the analysis, we have evaluated the base classes accuracy, novel classes accuracy, and the balance of both of them in each task of the simulation.
>
> - We have added a memory consumption analysis both for a client and server as presented in Appendix D.3. The server memory consumption is affected by the number of participating clients, while client memory consumption is affected by the batch-size value for local training. The server's memory consumption is relatively far smaller than the client's consumption. However, if the client has  low specification (resources),  the local training can be conducted by using a smaller batch size e.g. <= 16.
>
>
> W4.Although the model aims to be efficient, the use of advanced techniques like Neural ODEs could still demand significant computational resources, potentially limiting its applicability in resource-constrained environments. The paper does not sufficiently discuss this trade-off, which could be a critical consideration for practical deployment scenarios.
>
> Our Response: Thank you for your concern.
>
> - Our ablation study shows the performance of our method in several configurations. Configuration B shows the absence of dynamic prototypes as well as the rectification process. In such a configuration, the performance drops up to 4.6\%.
>
> - Our analysis of memory consumption as presented in Appendix D.3. shows that our method requires smaller memory consumption in the smaller batch-size. In the case of limited resources e.g. on edge devices, our method can be deployed with a small batch size. We also added running time analysis w.r.t. data size (simulated by percentage of available classes) and number of participating clients.

---

> ### Author Response · Authors · 2024-11-21
> **Author Response to Reviewer KvKJ for Q1-Q4**
>
> Q1. How does the scalability of the UOPP model hold up when there is a significant increase in the number of clients or data volume? Can the model adapt to the continuous addition of new categories without a significant increase in training time or resource consumption?
>
> Our Response: Thank you for your concern.
>
> - We have added memory analysis and run-time analysis for different participating local clients and class available class percentages as presented in Appendix X. Please note that the running time is calculated for a whole simulation where all the models and data are executed under a single GPU. In such a limitation (single GPU simulation for all clients) we are forced to run local training sequentially (not parallel). As a result, the running time is lear to the number of participating clients.  In the practical application, the running time must be much smaller since all the clients do the training in parallel across different devices.
>
> - Yes It can, As shown in section 6.3.f, our analysis on Complexity, Running Time, Parameters, and Communication Cost. The number of trainable and sharable parameters in the few-shot task remains the same i.e. 34M and 1.63MB respectively. The additional saved (in disk storage) parameter is the same as the shareable parameters i.e. 1.63 MB as it represents the prompt and prototypes for the current task.  Thus, our method can continuously learn new classes without a significant increase in resource consumption.
>
> Q2. How does the UOPP model perform in the face of class imbalance or distribution shift? Has the model undergone tests for generalization capabilities across different domains or tasks?
>
> Our Response: Thank you for your concern. We have added a new analysis for the performance of the consolidated methods in different non-i.i.d levels as presented in section 6.3.a. Our analysis shows that our method outperforms existing SOTAs in various non-i.i.d levels with a significant margin i.e. up to 40\%. Our further analysis in section 6.3.e shows the validation loss of novel classes with and without rectification. The analysis shows the rectification reduces the prototype bias significantly.
>
>
> Q3. What specific operations are used in the iterative correction process of dynamic prototypes to guide them closer to the actual data distribution? How is the number of iterations for prototype correction determined? Is there an early stopping mechanism to prevent overfitting?
>
> Our Response: Thank you for your concern.  In an iteration i.e. mini-batch, it conducts the prompt tuning and prototype rectification (adjustment) for M (40) steps. The adjusted prototype is produced by the GradNet g_{\psi}. The adjusted prototypes are then classified by the prototype-based head. The GradNet and the prompt are then updated by a single update process based on prototype cross-entropy loss. We explain the update procedure in section 4.1. and Appendix A.   The current version of our method doesn't utilize early stopping as each local training is conducted with small epochs. However, we have added an early stopping mechanism in our discussion for future study in Appendix E.
>
> Q4. Regarding the dynamic updating of prototypes, can Neural ODEs be replaced by other optimization strategies, or how significant is the impact on the results without the ODE method, and was its computational complexity considered?
>
>
> Our Response: Thank you for your concern.
>
> - We may utilize other rectification methods from Neural ODEs, but we can't ensure the its effectiveness. For example, a simple prototype adjustment (as utilized by PILoRA) can't work for this problem as shown by its poor performance on few-shot task. Neural ODEs have been proven significantly in our study. As shown in Configuration B of our ablation study, It shows the absence of dynamic prototypes and the rectification process (by Neural ODEs), drops the performance by up to 4.6\%.
>
> - The ablation analysis also shows the comparison of the simulation time of each configuration. The simulation time decreased by  33\% due to the absence of prototype rectification by Neural ODE. This is quite logical since rectification is conducted in each iteration (batch) and conducted in 40 steps.  Our analysis of memory consumption as presented in Appendix D.3, shows that our method works on various memory availability on the devices.

---

> > ### Comment · Reviewer_KvkJ · 2024-11-26
> > **The feedback was excellent, it resolved all my concerns.**
> >
> > The feedback was excellent, it resolved all my concerns, and I have no new worries. I have updated my score to 6, which is marginally above the acceptance threshold.

---

> > > ### Author Response · Authors · 2024-11-26
> > > **Author Response to Reviewer KvKJ**
> > >
> > > We appreciate Reviewer KvKJ for the time and effort to review our revised manuscript. We thank Reviewer KvKJ for increasing the score for our paper.

---

### Author Response · Authors · 2024-11-21
**Response to All Reviewers and Summary of Updates**

We express our thanks to all the reviewers for their constructive and insightful comments on our paper. We are honored that the reviewers (KvkJ, 2sKT, and 13iY) found our work to be novel and well-organized (b8Za, and 13iY).
We have revised our paper following the reviewers' feedback. We have improved the quality and presentation of our paper with the summary of changes as follows:

(1). We run PILoRA (ECCV 2024) which represents the current SOTA in FCIL as a new baseline for our method.

(2). We compared our method to the latest FSCIL SOTAs i.e. TEEN (NeurIPS 2023), NC-FSCIL(ICLR2023), OrCo(CVPR 2024), and PriViLege(CVPR 2024) on a standalone FSCIL setting emphasizing the effectiveness of our method to handle catastrophic forgetting with data scarcity problem. The newly added experiment and analysis are presented in Appendix D.1.

(3). We added a new experiment and analysis on different non-i.i.d. levels simulated by the percentage of available classes on each client.

(4). We added a new experiment and analysis on different k-shot values i.e. 3-shot (between 1 and 5) and 7-shot (more than 5), as presented in Appendix D2.

(5). We added a new measurement and analysis on memory consumption to answer the resource consumption that is related to the scalability, as presented in Appendix D.3.

(6). We added a new measurement and analysis on running time w.r.t. data size and number of local clients to address the scalability concern, as presented in Appendix D.4.

(7). We added a new discussion on limitations and potential solutions for future study and improvement, as presented in Appendix E.

(8). We elaborated our literature review both in the main paper and in the extended version (Appendix G) discussing the recent prototype-based methods.

(9).  We added missing details for symbols, settings, and procedures to improve the better understanding of our paper.

(10). Last but not least, we fixed the writing typos in our paper.

The changes w.r.t. feedback from reviewers 1(KvkL), 2(2sKT), 3(2VKj), 4(b8Za), and 5 (13iY) are highlighted in red, green, violet, orange and brown colors respectively.

---

### Author Response · Authors · 2024-12-04
**Summary of Revision and Discussion**

Dear Program Chairs (PC), Senior Area Chairs (SAC), Area Chairs (AC), and Reviewers,

As the discussion phase has ended,

**First**, we thank reviewers PC, SAC, AC, and Reviewers for your effort and time in organizing the review and discussion of our paper. Through these processes, we have improved the technical quality and presentation of our paper significantly. We thank the reviewers for constructive feedback on our initial paper, our revised paper, and responsive discussion confirming the novelty and significance of our works. We appreciate reviewer 13iY, KvKJ, and b8Za for maintaining and increasing their score for our paper.

**Second**, we have revised our paper following the reviewers' advice in both writing i.e. **layout, presentation, details, symbols, captions,  spacing, typo-fixing, etc**, and technical aspects i.e. **stronger motivation, problem formulation, uniqueness of proposed method, metrics, performance on different non-iid levels and k-shots, new FCIL SOTA comparator (PILoRA), comparison with recent FSCIL SOTAs in standalone FSCIL, memory analysis, detailed running time analysis, discussion on limitation and potential solution, and detailed setting and literature study, etc** as we mentioned in our previous official comment.

**Third**, we would like to confirm that we have addressed all concerns and questions of the reviewers in our revised paper and responses. In addition, we would like to highlight a key point that may not have been reviewed by reviewers 2sKT in the discussion phase, as follows:

- **Data Leakage and Pseudo-Rehearsal issues**:  ViT is a non-reversed function, and prototype space is different from input space both in terms of format and data ranges, thus we can't directly generate input image by inputting prototype into ViT model (Section 4.2: How to generate prototypes). Thus prorotype sharing doesn't breach data privacy policy.  Our method complies with non-rehearsal and non-pseudo-rehearsal policy since $\mathcal{Q}$ is drawn from $\hat{Z}_l$ only, not the whole  $Z_l$ (Section 4.3, last paragraph).

I think that's all, once again thank you to all the committee, we pray that the hard work and dedication of the committee will be paid off by the success and impact of this year's ICLR.

Best Regards,

Author of Submission 6715

---

### Meta-Review · Area_Chair_gVHc · 2024-12-15

**Metareview:**

The paper proposes a Federated Few-Shot Class-Incremental Learning (FFSCIL) framework and introduces the Unified Optimized Prototype Prompt (UOPP) model to address challenges like catastrophic forgetting, overfitting, and prototype bias. While the paper’s problem formulation is interesting, reviewers raised concerns regarding the clarity of its motivations, the uniqueness of the proposed methods, and the completeness of comparisons with related baselines.

The authors made significant revisions, including clarifying motivations, improving writing, and addressing reviewers' concerns about data privacy and comparisons. After the author-reviewer discussion period, most of the reviewers confirmed that the authors had addressed their concerns and gave positive final ratings. However, Reviewer 2sKT still has concerns about the novelty.

Considering that most of the reviewers gave positive ratings on this paper, I recommend acceptance. However, Reviewer 2sKT's comments and concerns should be summarized and included in the final version.

**Additional Comments On Reviewer Discussion:**

During the rebuttal period, reviewers raised concerns about the novelty and motivation of FFSCIL, the comparisons with related baselines, and potential data privacy issues in the use of prototypes. The authors responded comprehensively, adding comparisons to recent baselines (e.g., PILoRA), revising the metrics (e.g., harmonic mean accuracy), and clarifying the privacy-preserving aspects of their method. Despite these efforts, Reviewer 2sKT maintained concerns about the novelty of the approach.

Considering that most of the reviewers gave positive ratings on this paper, I recommend acceptance. However, Reviewer 2sKT's comments and concerns should be summarized and included in the final version.

---

### Decision · Program_Chairs · 2025-01-22

Accept (Poster)